# Diagnosing the Reliability of LLM-as-a-Judge via Item Response Theory

**Junhyuk Choi** [1] [*]   **Sohhyung Park** [2] [*]   **Chanhee Cho** [1]   **Hyeonchu Park** [1]   **Bugeun Kim** [1]

## Abstract

While LLM-as-a-Judge is widely used in automated evaluation, existing validation practices primarily operate at the level of observed outputs, offering limited insight into whether LLM judges themselves function as stable and reliable measurement instruments. To address this limitation, we introduce a two-phase diagnostic framework for assessing reliability of LLM-as-a-Judge, grounded in Item Response Theory (IRT). The framework adopts Graded Response Model (GRM) of IRT and formalizes reliability along two complementary dimensions: (1) intrinsic consistency, defined as the stability of measurement behavior under prompt variations, and (2) human alignment, capturing correspondence with human quality assessments. We empirically examine seven LLM judges with this framework, and show that leveraging IRT-GRM yields interpretable signals for diagnosing judgments systematically. These signals provide practical guidance for verifying reliability of LLM-as-a-Judge and identifying potential causes of unreliability. Code: github.com/elu-lab/IRT-Judge

## 1. Introduction

Large Language Models (LLMs) have emerged as automated evaluators across diverse domains, commonly referred to as LLM-as-a-Judge (Liu et al., 2023; Gu et al., 2024; Li et al., 2024). This paradigm has been rapidly adopted not only in natural language processing tasks such as summarization (Fabbri et al., 2021; Crupi et al., 2025; Gao et al., 2023) and dialogue evaluation (Mehri & Eskenazi, 2020; Chan et al., 2024), but also extends to vision-language models (Ku et al., 2024a; Chen et al., 2024; Lee

et al., 2024), and reward modeling for reinforcement learning from human feedback (Wang et al., 2024c; Xu et al., 2025b). The appeal is clear: LLM judges offer scalable, cost-effective evaluation without the bottleneck of human annotation. Despite their widespread adoption, a fundamental question remains unresolved:

*Can we trust the judgments of LLM-as-a-Judge?*

LLM judges are increasingly used to support critical evaluation decisions, even though principled methods for verifying the reliability of their judgments remain limited. The reliability of LLM-based evaluation can be examined through two fundamental dimensions: ***intrinsic consistency***, which concerns whether a judge behaves as a stable and coherent measurement instrument under equivalent evaluation conditions, and ***human alignment***, which concerns how closely its assessments correspond to those of human evaluators.

However, prior work tends to address two dimensions separately and relies on metrics that provide limited diagnostic insight. Studies focusing on intrinsic consistency have proposed a range of approaches, including inter-rater agreement (Kollitsch et al., 2024; Wang et al., 2024a), internal consistency (Schroeder & Wood-Doughty, 2025), and uncertainty quantification (Wagner et al., 2024; Xiong et al., 2024). While these methods capture partial aspects of judge behavior, they operate at the level of observed scores and cannot separate stable measurement characteristics of the judge from variation driven by the evaluated samples. Alignment, which has been widely studied in prior work, is commonly assessed through aggregate agreement signals between LLM and human judgments, typically quantified using correlation and agreement based metrics (Gu et al., 2024; Liu et al., 2023). However, agreement at the outcome level reflects similarity in evaluation results, rather than whether the judge itself functions as a reliable measurement instrument. As a result, the reliability of LLM judges remains insufficiently characterized under existing evaluation approaches.

To address this gap, we introduce a unified framework for assessing LLM judge reliability that integrates intrinsic consistency and human alignment into a single diagnostic view. We leverage Item Response Theory (IRT; Embretson & Reise (2013)), a measurement framework widely used to analyze latent traits from observed responses, to examine LLM judging behavior. IRT models observed scores as probabilis-

---

[*]Equal contribution   [1]Department of Artificial Intelligence, Chung-Ang University, Seoul, Republic of Korea [2]Department of Industrial Engineering, Seoul National University, Seoul, Republic of Korea. Correspondence to: Junhyuk Choi <chlwnsgur129@cau.ac.kr>, Bugeun Kim <bgnkim@cau.ac.kr>.

*Proceedings of the 43rd International Conference on Machine Learning*, Seoul, South Korea. PMLR 306, 2026. Copyright 2026 by the author(s).

tic functions of latent traits and item characteristics, enabling judgments to be analyzed in terms of underlying evaluation signals rather than outcome-level scores alone (Lord, 1952; Embretson & Reise, 2013). For rating-based judgments with ordered categorical outputs (Likert-style ratings), we adopt the Graded Response Model (GRM; Samejima (1969)), a polytomous IRT model that characterizes how latent evaluation signals give rise to discrete score categories. Just as educational testing uses IRT to assess whether exam items reliably measure student ability, we use GRM to assess whether LLM judges reliably measure subject quality.

Our diagnostic framework is organized into two conceptual phases. The first phase examines *intrinsic consistency* by applying prompt perturbations and examining whether judges produce consistent measurements under semantically equivalent variations. Phase 2 evaluates *human alignment* by comparing latent evaluation signals inferred from LLM judges with those of human annotators, conditioned on judges that exhibit sufficient intrinsic consistency. This sequential design ensures that alignment is interpreted only when measurement behavior is internally consistent.

We evaluate the framework across diverse evaluation settings, illustrating how IRT-GRM produces interpretable diagnostic signals that characterize measurement behavior of LLM judges. By identifying multiple sources of unreliability, the framework provides a practical basis for determining when LLM-based evaluation can be considered reliable.

## 2. Related Work

LLM-as-a-Judge is now widely used as a scalable alternative or complement to human evaluation (Chiang & Lee, 2023; Gu et al., 2024; Li et al., 2024; Tan et al., 2024), with recent work further systematizing this paradigm via structured design choices including Chain-of-Thought prompting (Wei et al., 2022) to elicit explicit reasoning (Wang et al., 2024b; Wagner et al., 2024; Chiang et al., 2025; Saha et al., 2025), checklist-based evaluation (Furuhashi et al., 2025), external validation tools (Findeis et al., 2025), self-improvement processes (Wu et al., 2025), and agent-based pipelines (Zhuge et al., 2025). Despite its growing adoption, concerns have been raised about whether LLM-based evaluation can produce judgments that are sufficiently reliable and consistent to substitute for human evaluation (Chehbouni et al., 2025; Bavaresco et al., 2025). Yet existing evidence offers limited guidance on whether failures in LLM judging stem from unreliable scoring or divergence from human judgment.

**Intrinsic Consistency** Prior work has assessed consistency by examining whether an LLM judge's outcomes remain stable across different inputs and evaluation settings. These include measuring inter-rater agreement across LLM evaluators (Kollitsch et al., 2024; Wang et al., 2024a), as

well as evaluating internal consistency by repeating judgments under different random seeds (Patel et al., 2024) and summarizing stability with metrics such as McDonald's $\omega$ (Schroeder & Wood-Doughty, 2025). However, relying only on final scores or discrete verdicts provides limited insight into the stability and confidence of LLM judgments, and recent work has incorporated uncertainty modeling (Xu et al., 2024; Yona et al., 2024; Xiong et al., 2024), including leveraging token probabilities associated with ratings or verdicts to quantify evaluation confidence or uncertainty (Wagner et al., 2024; Xie et al., 2025; Sheng et al., 2025). In addition, several works have examined multiple dimensions of reliability in LLM-based evaluation, including structural instability (Xu et al., 2025a), rating indeterminacy (Guerdan et al., 2025), and biases that affect judging outcomes (Ye et al., 2025; Shi et al., 2025; Thakur et al., 2025b). However, these approaches capture only partial aspects of reliability without separating the judge's measurement properties from sample quality, leaving researchers unable to diagnose the source of observed inconsistencies in measurements.

**Human Alignment** Human alignment captures the extent to which LLM-based judgments agree with human evaluations, often serving as a key objective of LLM-as-a-Judge benchmarks and evaluation frameworks (Liu et al., 2023; Zheng et al., 2023). It is commonly measured by comparing LLM-based and human evaluations using correlation-based metrics (Kendall's $\tau$, Spearman's $\rho$, or Pearson's $r$) (Xu et al., 2025a; Gera et al., 2025; Chiang et al., 2025; Bavaresco et al., 2025) and agreement metrics, including percent agreement, pairwise accuracy, Cohen's $\kappa$ (Cohen, 1960), Krippendorff's $\alpha$ (Hayes & Krippendorff, 2007), or Scott's $\pi$ (Scott, 1955; Furuhashi et al., 2025; Guerdan et al., 2025; Bavaresco et al., 2025; Zhang et al., 2025; Thakur et al., 2025a; Haldar & Hockenmaier, 2025). However, relying on single-point scores or discrete verdict matches may provide an incomplete picture of human alignment, as such summaries can mask disagreement and uncertainty in human evaluation. To address this, recent work explores distributional alignment that predicts human rating distributions (Chen et al., 2025) and statistical frameworks that analyze discrepancies at the distribution level (Polo et al., 2025). Moreover, recent work proposes procedural approaches to justify or guarantee human agreement, including selective evaluation with escalation (Jung et al., 2025) and statistical tests for replacing human annotators with LLM judges (Calderon et al., 2025). Yet these metrics offer limited insight into the nature of misalignment, unable to distinguish inconsistent measurements or apply different circumstances.

## 3. Methodology

We propose a framework for diagnosing LLM-as-a-judge reliability based on Item Response Theory (IRT). Our ap-

proach consists of three components: (1) fitting a Graded Response Model (GRM) to separate measurement properties from true quality differences, (2) generating controlled prompt variations to probe sensitivity, and (3) extracting interpretable reliability metrics grounded in social science.

### 3.1. Graded Response Model for LLM Judges

We adopt the GRM (Samejima, 1969) to jointly estimate sample quality and measurement properties. For a rating scale with $K$ unique score values, the probability that an evaluator with a variation of judgment prompt $p \in \mathcal{P}$ associates score value $Y_{pj}$ of evaluation subject $j \in \mathcal{D}$ (instance to be evaluated) with at least $k \in \{1, 2, \cdots, K\}$ is:

$$P(Y_{pj} \geq k \mid \theta_j) = \frac{1}{1 + \exp(-\alpha_p(\theta_j - \beta_{pk}))} \quad (1)$$

where $\theta_j \in \mathbb{R}$ is the latent genuine quality of subject $j$, $\alpha_p \in \mathbb{R}^+$ is the discrimination parameter for the variation $p$, and $\{\beta_{pk}\}_{k=1}^K \in \mathbb{R}$ is a monotonically increasing sequence of thresholds defining levels of latent quality at which the judge transitions between adjacent score values. After fitting GRM, each prompt variant (original, typo, newline, paraphrase) corresponds to its own $(\alpha_p, \boldsymbol{\beta}_p)$, while all variants share the latent quality $\theta_j$ per sample. This separation allows us to attribute score variation to either prompt effects (captured by $\alpha, \beta$) or true quality differences ($\theta$).

Also, to follow the practice of GRM, we apply priors $\theta_j \sim \mathcal{N}(0, 1)$, $\alpha_p \sim \text{LogNormal}(0, 0.5)$, and $\beta_{pk} \sim \mathcal{N}(0, 1)$ with an ordered constraint ($\beta_{pk} < \beta_{p,k+1}$ for all $k < K$). We perform posterior inference via No-U-Turn Sampler (NUTS; Hoffman et al. (2014)) with multiple chains and samples. We adopt GRM as the standard IRT model for ordered categorical responses, matching our Likert-scale setting. Alternative models are unsuitable. NRM (Bock, 1972) assumes nominal categories, and PCM (Masters, 1982) assumes uniform discrimination across items, restrictive when prompts exhibit varying sensitivity. A latent-variable derivation of Eq. (1) and a discussion of why $\theta_j$ is identifiable and comparable across judges are provided in Appendix A.

### 3.2. Why Should We Focus on Latent Quality ($\theta$)?

A critical challenge in comparing LLM judges is that different models use the output range of rating differently. When instructed to use a 5-point scale, one model might produce all five score values while another effectively answers only three values. This inconsistency renders direct comparison of fitted model ($\alpha$ and $\beta$) problematic.

Consider two judges evaluating the same samples on a 5-point scale: Judge A mainly produced scores $\{1, 2, 3, 4, 5\}$, while Judge B answered only $\{3, 4, 5\}$. Because fitting GRM requires at least one sample for each rating value, fitting on A's result yields four threshold parameters. But,

fitting on B's result yields two threshold parameters due to the absence of rating data for 1 and 2. Thus, the information on the observed ratings might be overrepresented in $\alpha$ and $\beta$, which prohibits observing general tendency; we cannot ensure whether the same $\alpha$ and $\beta$ are applicable to the missing rating values. Hence, we considered using $\theta$.

In contrast, latent quality ($\theta$) represents an intrinsic property of the evaluation subject $j$, not a property of the measurement instrument. As shown in equation 1, $\theta_j$ is shared across different rating values. So, regardless of whether the judge actually produced a specific rating value or not, distribution of $\theta_j$ can hint at the judgment behavior of a judge. This invariance makes $\theta_j$ the appropriate basis for comparing different judgment methods, prompts, or models.

### 3.3. Prompt Variation Design

To assess prompt sensitivity, we apply three minimal perturbations that preserve semantic content: (1) *typo* introduces character-level errors in high-attention words, (2) *newline* inserts sentence-level line breaks, and (3) *paraphrase* substitutes synonyms for verbs and adjectives. These reflect common real-world variations arising from trivial edits; if a judge's scores fluctuate substantially under such minor changes, this signals a reliability concern independent of alignment with human ratings. Importantly, Phase 1 evaluates the stability of a single fixed prompt against surface-level noise, not comparisons across structurally different prompts. Full-sentence paraphrasing or structural reformulation (reordering rubric items, changing evaluation framing) substantially alter LLM outputs (Sclar et al., 2024; Mizrahi et al., 2024; Cox et al., 2025) and effectively constitute a new measurement instrument, which should undergo Phase 1 independently. See Appendix B for further discussion.

### 3.4. Reliability Metrics

Our framework extracts four metrics organized into two diagnostic phases. In Phase 1, we assess whether the LLM judge functions as a reliable measurement instrument without requiring human reference. In Phase 2, applicable only after Phase 1 criteria are satisfied, we examine how the judge's quality perception differs from human judgment.

#### 3.4.1. PHASE 1: INTRINSIC CONSISTENCY

Before comparing an LLM judge to humans, we must first establish whether it functions as a reliable measurement. The following two metrics address this question.

**Prompt Consistency** ($C_V$). A reliable judge should produce consistent evaluations regardless of prompt variations. We quantify this using the within-rating variance of latent quality estimates. For each prompt variant $p$, we compute variance of $\theta_j$ within each rating value and take an average

across those variances, denoting as $\bar{V}_p$:

$$\bar{V}_p = \frac{1}{|K_p| - 1} \sum_{k \in K_p} \mathrm{Var}_{j \sim \mathcal{D}_p(k)}(\theta_j), \quad (2)$$

$$\mathcal{D}_p(k) = \{j \mid j \in \mathcal{D} \wedge Y_{pj} = k\},$$

$$K_p = \{k \mid D_p(k) \neq \emptyset\}.$$

We then compute the coefficient of variation across prompts:

$$C_V = \frac{\sigma_V}{\mu_V} \quad (3)$$

where $\sigma_V$ and $\mu_V$ are the standard deviation and the average of $\bar{V}_p$ across prompts. Note that as lower $\bar{V}_p$ indicates tighter clustering of samples within each rating value, lower $C_V$ means more consistent measurement. We regard $C_V < 0.1$ as acceptable consistency, where variance is less than 10%. This threshold is grounded in established conventions[1].

**Marginal Reliability ($\rho$).** This metric, standard in psychometric practice (Embretson & Reise, 2013), quantifies how much variance in estimated $\theta$ reflects true quality differences versus estimation uncertainty:

$$\rho = \frac{\mathrm{Var}(\hat{\theta}_j)}{\mathrm{Var}(\hat{\theta}_j) + \mathbb{E}[\sigma_j^2]} \quad (4)$$

where $\hat{\theta}_j$ and $\sigma_j^2$ denotes posterior mean and variance across NUTS samples drawn from each $\theta_j$. Here, the variance of such sample mean $\mathrm{Var}(\hat{\theta}_j)$ represents true variation of latent quality, while the expected variation of $\sigma_j^2$ indicates estimation uncertainty during NUTS sampling. Thus, low $\rho$ suggests the model is more prone to measurement errors and lacks fundamental capability to serve as a judge. For example, a value of $\rho = 0.7$ means 70% of variance reflects true quality differences while 30% is measurement error. We establish interpretation guideline following psychometric conventions; $\rho > 0.7$ indicates acceptable reliability following psychometric conventions (Nunnally, 1975).

**Diagnostic Interpretation.** The combination of $C_V$ and $\rho$ enables differential diagnosis: high $C_V$ with acceptable $\rho$ points to prompt sensitivity as the primary issue, while low $\rho$ regardless of $C_V$ indicates fundamental model limitations.

### 3.4.2. PHASE 2: HUMAN ALIGNMENT

Once a judge passes Phase 1 criteria, we examine how its quality perception compares to human judgment. Two metrics capture different aspects of this comparison. As we only

[1]By Chebyshev's inequality (Saw et al., 1984), $C_V < 0.1$ guarantees that at least 75% of assessments lie within $\pm 20\%$ of the mean under any distribution, and it is widely adopted as a benchmark for high measurement precision in analytical chemistry (Reed et al., 2002) and clinical epidemiology (Shechtman, 2013).

use original prompts instead of using all variants, we omit the index $p$ in the following equations for simplicity.

**Discrimination Breadth Ratio ($\theta_{\mathbf{ratio}}$).** We compare the range of latent quality estimates between the LLM judge and human annotators:

$$\theta_{\mathrm{ratio}} = \theta_{\mathrm{range}}^{(\mathrm{LLM})} / \theta_{\mathrm{range}}^{(\mathrm{Human})} \quad (5)$$

$$\theta_{\mathrm{range}} = \mathrm{median}(\theta_j \mid Y_j = \max Y_{j'})$$
$$- \mathrm{median}(\theta_j \mid Y_j = \min Y_{j'}) \quad (6)$$

This ratio reveals whether the LLM discriminates quality differences with similar breadth to humans. A ratio less than 1 indicates the LLM perceives a narrower quality spectrum—it is *hypersensitive*, failing to distinguish samples that humans would rate differently. A ratio greater than 1 indicates the LLM perceives a wider spectrum—it is *insensitive*, amplifying quality differences beyond human perception. A ratio near 1 suggests comparable discrimination breadth. Note that $\theta_{\mathrm{range}}$ alone is uninterpretable without human reference: a value of 1.5 could indicate appropriate, insufficient, or excessive discrimination depending on the human baseline.

**Human Alignment ($D_W$).** To assess absolute alignment with human judgments, we compute the Wasserstein distance between LLM and human distributions of $\hat{\theta}_j$:

$$D_W = W_1(\hat{\theta}^{(\mathrm{LLM})}, \hat{\theta}^{(\mathrm{Human})}) \quad (7)$$

We choose Wasserstein over alternatives because correlation captures only linear relationships ignoring distributional shape, and KL divergence is asymmetric and undefined for non-overlapping support. Wasserstein captures both location shift and distributional differences, providing an interpretable cost of transforming one distribution into another.

**Diagnostic Interpretation.** The combination of $\theta_{\mathrm{ratio}}$ and $D_W$ enables nuanced comparison: $\theta_{\mathrm{ratio}} \approx 1$ with high $D_W$ indicates the judge discriminates with appropriate breadth but is systematically more lenient or strict, while $\theta_{\mathrm{ratio}} \neq 1$ with low $D_W$ suggests the judge's overall perception aligns with humans despite differing sensitivity. Both metrics deviating from ideal values indicates fundamental differences in how the LLM perceives quality.

## 4. Demonstration

### 4.1. Judge prompts and Benchmarks

We select Judge methods that provide rating-based evaluations across diverse criteria. All methods were evaluated on human annotations, enabling intrinsic consistency diagnostics (Phase 1). For detailed description, refer to Appendix C.

**NLP.** We evaluate two LLM-as-a-judge methods: G-Eval (Liu et al., 2023) and HelpSteer-2 (Wang et al., 2024c) to mirror popular usages of LLM-based quality evaluation. We evaluated reliability of G-Eval on two datasets: SummEval (Fabbri et al., 2021) and TopicalChat (Mehri & Eskenazi, 2020). For SummEval, we use the original human annotation script from the dataset, measuring summary quality with four 5-point scales. For TopicalChat, we adapt the SummEval format with metric definitions from Mehri & Eskenazi (2020), evaluating dialogue quality with two binary scales (understandability, groundedness) and three 3-point scales (naturalness, coherence, engagingness). For HelpSteer-2, we use the original prompt from Wang et al. (2024c), measuring response quality with five 5-point scales.

**Vision.** We evaluate one VLM-as-a-judge method: VIEScore (Ku et al., 2024a) to mirror popular usages of image quality evaluation. We follow the same prompt on the same dataset. We evaluated reliability of VIEScore on ImageHub dataset (Ku et al., 2024b) containing three subsets: control-guided image generation (CIG), text-guided image editing (TIE), and mask-guided image editing (MIE). VIEScore measures image-text alignment with two 11-point scales: semantic consistency (SC) and perceptual quality (PQ). Thus, we separately examined VIEScore on these subsets.

### 4.2. Selected Models

We select judge models based on two criteria: (1) frontier models widely adopted in LLM-as-a-Judge deployments, and (2) coverage across modalities to enable cross-modal reliability comparison across diverse benchmarks. All evaluations are conducted via the OpenRouter API (OpenRouter, 2025) with temperature set to 0 for deterministic outputs. Specifically, we used seven models from four families: Gemini 2.5 Flash (Comanici et al., 2025), GPT-4o and GPT-4o-mini (Hurst et al., 2024), Qwen3-30B-A3B-instruct and Qwen3-235B-A22B-instruct (Yang et al., 2025), and LLaMA-4-Maverick and LLaMA-4-Scout (Meta AI, 2025). Note that, for vision tasks, we used Qwen3-VL (Bai et al., 2025) instead of Qwen3 to ensure text and vision inputs.

### 4.3. Procedure for Analysis

**Step 1. Generating Prompt Variations.** We generate three prompt variations for each evaluation prompt: typo, newline, and paraphrase. For *typo* variations, we select the five high-attention tokens[2] from the final layer of Qwen3-8B (Yang et al., 2025) and apply character-level perturbations to those tokens using the AugLy library (Papakipos & Bitton, 2022). For *newline* variations, we randomly add three newlines between sentence segments. For *paraphrase* variations, we extract five verbs or adjectives using NLTK POS

---

[2]After excluding metric names, we chose five highest words.

tagging (Bird & Loper, 2004), then use GPT-4o-mini to generate synonyms for those tokens. To ensure comparability across tasks, identical variations are applied to all models; details are provided in Appendix B.

**Step 2. Fitting IRT-GRM.** We fit GRM using PyMC (Salvatier et al., 2016) with the NUTS sampler (Hoffman et al., 2014) via NumPyro (Phan et al., 2019) backend. We run 4 chains with 1000 warmup and 1000 sampling iterations, setting target acceptance rate to 0.95. For binary scale, we fit a 2-parameter logistic model instead as GRM assumes $K > 2$. Detailed hyperparameters are provided in Appendix A.

## 5. Results 1: Intrinsic Consistency

Table 1 presents the Phase 1 diagnostic results, where we highlight results meeting consistency criteria: $C_V \leq 0.10$ and $\rho \geq 0.7$. With two metrics, we analyze LLM-as-a-judge methods along three axes: modality, model, and task.

### 5.1. Prompt Consistency ($C_V$)

**Modality: VIEScore subtasks exhibit higher prompt sensitivity than the NLP subtasks.** NLP judges on SummEval, TopicalChat, or HelpSteer-2 generally maintain $C_V < 0.30$, with several model-criterion pairs achieving $C_V < 0.10$. In contrast, VIEScore exhibits substantially higher $C_V$, ranging from 0.16 to 1.32. This gap is most pronounced for Gemini-2.5, which showed score from 0.03 to 0.29 on NLP tasks but exceeds $C_V > 1.0$ across all VIEScore subtasks. These results suggest that vision-language evaluation is inherently more sensitive to prompt wording than text-based evaluation.

**Model: Scale improves prompt robustness in NLP subtasks but not in VIEScore.** In NLP benchmarks, larger models consistently achieve lower $C_V$. Qwen3-235B outperforms Qwen3-30B on SummEval (Relevance: 0.09 vs. 0.17, Fluency: 0.09 vs. 0.15, Coherence: 0.16 vs. 0.22). GPT-4o similarly surpasses GPT-4o-mini (SummEval Fluency: 0.06 vs. 0.60; HelpSteer-2 Helpfulness: 0.08 vs. 0.30). However, this scaling effect does not transfer to VIEScore: in VIEScore, Qwen3-30B and Qwen3-235B show comparable $C_V$, and GPT-4o-mini occasionally matches or exceeds GPT-4o. This modality-dependent scaling suggests that prompt robustness in vision-language evaluation may rely on capabilities distinct from those improved by scale.

**Task: Summarization is most stable; VIEScore is not.** Within NLP, judgment criteria in SummEval achieves the lowest $C_V$ overall, with most models maintaining $C_V < 0.20$. Those in TopicalChat and HelpSteer-2 show moderate variability, though HelpSteer-2 Complexity exhibits notable instability for Gemini-2.5 (1.08). VIEScore consistently

*Table 1.* Phase 1 intrinsic consistency results. CV denotes prompt consistency; $\rho$ denotes marginal reliability (higher is better). Highlighted cells indicate CV $\leq 0.10$ and $\rho \geq 0.70$. Gemini-2.5 refers to Gemini 2.5 Flash; Llama-4-m and Llama-4-s refer to Llama-4-Maverick and Llama-4-Scout, respectively. For VIEScore, Qwen3-30B and Qwen3-235B refer to their VL variants

| Benchmark | | Gemini-2.5 | | GPT-4o | | GPT-4o-mini | | Qwen3-30b | | Qwen3-235b | | Llama-4-m | | Llama-4-s | |
|---|---|---|---|---|---|---|---|---|---|---|---|---|---|---|---|
| | | $C_V$ | $\rho$ | $C_V$ | $\rho$ | $C_V$ | $\rho$ | $C_V$ | $\rho$ | $C_V$ | $\rho$ | $C_V$ | $\rho$ | $C_V$ | $\rho$ |
| **SummEval** | *Relevance* | 0.25 | 0.91 | **0.05** | **0.92** | **0.04** | **0.93** | 0.17 | 0.86 | **0.09** | **0.89** | 0.36 | 0.83 | 0.17 | 0.92 |
| | *Consistency* | 0.07 | 0.55 | **0.05** | **0.91** | 0.92 | 0.88 | 0.15 | 0.60 | 0.08 | 0.66 | 0.25 | 0.63 | 0.07 | 0.78 |
| | *Fluency* | 0.21 | 0.89 | **0.06** | **0.90** | 0.60 | 0.89 | 0.15 | 0.90 | **0.09** | **0.91** | 0.13 | 0.81 | 0.15 | 0.92 |
| | *Coherence* | **0.09** | **0.89** | 0.29 | 0.94 | **0.06** | **0.92** | 0.22 | 0.87 | 0.16 | 0.92 | 0.18 | 0.84 | 0.15 | 0.88 |
| **TopicalChat** | *Understandability* | 0.18 | 0.39 | 0.16 | 0.48 | 0.20 | 0.49 | 0.03 | 0.53 | 0.27 | 0.34 | 0.21 | 0.46 | 0.48 | 0.43 |
| | *Naturalness* | 0.29 | 0.85 | 0.28 | 0.57 | 0.11 | 0.80 | 0.18 | 0.87 | 0.23 | 0.87 | 0.49 | 0.71 | 0.33 | 0.81 |
| | *Coherence* | 0.27 | 0.79 | 0.14 | 0.84 | 0.17 | 0.81 | 0.13 | 0.85 | 0.17 | 0.86 | 0.25 | 0.81 | 0.33 | 0.82 |
| | *Engagingness* | 0.15 | 0.78 | 0.52 | 0.72 | 0.20 | 0.79 | 0.21 | 0.87 | 0.16 | 0.86 | 0.21 | 0.70 | 0.11 | 0.80 |
| | *Groundedness* | 0.17 | 0.72 | **0.07** | **0.71** | 0.12 | 0.70 | 0.18 | 0.73 | 0.30 | 0.71 | **0.10** | **0.71** | 0.18 | 0.73 |
| **HelpSteer-2** | *Helpfulness* | **0.03** | **0.86** | **0.08** | **0.84** | 0.30 | 0.67 | 0.27 | 0.72 | 0.37 | 0.78 | 0.13 | 0.64 | 0.28 | 0.66 |
| | *Correctness* | **0.10** | **0.72** | 0.11 | 0.76 | 0.12 | 0.60 | 0.30 | 0.68 | 0.25 | 0.76 | 0.24 | 0.54 | 0.28 | 0.54 |
| | *Coherence* | 0.16 | 0.68 | 0.24 | 0.67 | 0.40 | 0.54 | 0.21 | 0.58 | 0.78 | 0.60 | 0.25 | 0.40 | 0.31 | 0.49 |
| | *Complexity* | 1.08 | 0.87 | 0.16 | 0.89 | 0.30 | 0.81 | 0.39 | 0.88 | 0.32 | 0.89 | 0.25 | 0.77 | 0.31 | 0.85 |
| | *Verbosity* | 0.17 | 0.87 | 0.04 | 0.83 | 0.21 | 0.74 | 0.20 | 0.76 | 0.74 | 0.76 | 0.33 | 0.75 | 0.45 | 0.83 |
| **VIEScore** | *CIG-SC* | 1.32 | 0.94 | 0.46 | 0.93 | 0.56 | 0.93 | 0.32 | 0.94 | 0.37 | 0.92 | 0.36 | 0.90 | 0.29 | 0.81 |
| | *PQ* | 1.11 | 0.94 | 0.30 | 0.96 | 0.82 | 0.93 | 0.62 | 0.94 | 0.47 | 0.94 | 0.17 | 0.92 | 0.46 | 0.83 |
| | *MIE-SC* | 1.01 | 0.94 | 0.37 | 0.95 | 0.52 | 0.94 | 0.45 | 0.94 | 0.94 | 0.95 | 0.54 | 0.93 | 0.55 | 0.87 |
| | *PQ* | 1.14 | 0.92 | 1.02 | 0.94 | 0.58 | 0.93 | 0.40 | 0.94 | 0.61 | 0.90 | 0.32 | 0.89 | 0.58 | 0.80 |
| | *TIE-SC* | 1.00 | 0.94 | 0.32 | 0.94 | 1.11 | 0.95 | 0.64 | 0.93 | 0.43 | 0.95 | 0.56 | 0.93 | 0.80 | 0.88 |
| | *PQ* | 1.14 | 0.93 | 0.68 | 0.92 | 0.30 | 0.93 | 0.16 | 0.92 | 0.67 | 0.91 | 0.84 | 0.90 | 0.77 | 0.80 |

yields high $C_V$ across all models and subtasks, indicating that prompt sensitivity is a task-level characteristic rather than a model-specific limitation.

### 5.2. Marginal Reliability ($\rho$)

**Modality: VIEScore achieves high marginal reliability.** Despite high prompt sensitivity, VIEScore achieves the highest $\rho$ values among all benchmarks (0.80–0.96). This apparent paradox suggests that while vision-language judges are sensitive to exact prompt wording, they produce reliable quality orderings once a specific prompt is fixed. NLP benchmarks show more variance: SummEval achieves consistently high $\rho$ (0.81–0.94), while TopicalChat and HelpSteer-2 exhibit criterion-dependent reliability.

**Model: scale impacts on NLP but not on VIEScore.** In NLP benchmarks, larger models consistently achieve higher $\rho$. Qwen3-235B outperforms Qwen3-30B on SummEval (0.89–0.92 vs. 0.86–0.90) and HelpSteer-2 Helpfulness (0.78 vs. 0.72) and Correctness (0.76 vs. 0.68). Similarly, GPT-4o surpasses GPT-4o-mini on HelpSteer-2 across all criteria (e.g., Helpfulness: 0.84 vs. 0.67, Correctness: 0.76 vs. 0.60). However, this scaling effect does not transfer to vision tasks: in VIEScore, Qwen3-30B and Qwen3-235B achieve comparable $\rho$ (0.90–0.95), and GPT-4o-mini occasionally matches or surpasses GPT-4o. This modality-dependent scaling implies that vision-language evaluation may rely on capabilities which do not scale with model size.

**Task: TopicalChat Understandability is least reliable.** In NLP, TopicalChat Understandability consistently yields

the lowest $\rho$ across all models (0.34–0.53), substantially below the conventional threshold of $\rho > 0.7$. Other TopicalChat criteria (Naturalness, Coherence, Engagingness) achieve moderate-to-high reliability (0.70–0.87), while Groundedness remains borderline (0.70–0.73). HelpSteer-2 shows criterion-dependent patterns: Helpfulness and Complexity achieve high $\rho$ (0.66–0.89), while Coherence underperforms (0.40–0.68). Criteria in SummEval maintains consistently high reliability across all criteria.

### 5.3. Discussion

Phase 1 suggests that consistency of current LLM-as-a-judge criteria might depend on how the prompt specifies evaluation criteria. As shown in Table 1, there is no free lunch; no model showed acceptable consistency across all criteria when we quantified $C_V$ and $\rho$. Thus, one should verify the effect of prompts and models when suggesting new LLM judgment tasks.

It is still questionable why such difference occur. To diagnose the cause, we conduct two simple ablation studies on evaluation criteria, using TopicalChat[3]. First, we test whether detailed evaluation instructions or chain-of-thought prompting improve consistency ($C_V$). Second, we examine whether the rating scales used in existing criteria are adequate for reliable judging ($\rho$) by comparing the original 3-point scale with alternative scales: 5-point and 7-point.

The result of ablation study suggests proper way to design judgment prompts, as shown in Table 3. First, instruc-

---

[3]For the result on VIEScore, see Appendix D

*Table 2.* Phase 2 human alignment results. $\theta_{ratio}$ denotes discrimination breadth ratio; $D_W$ denotes Wasserstein distance between LLM and human $\theta$ distributions. Gemini-2.5 refers to Gemini 2.5 Flash; Llama-4-m and Llama-4-s refer to Llama-4-Maverick and Llama-4-Scout, respectively. For VIEScore, Qwen3-30B and Qwen3-235B refer to their VL variants.

| Benchmark | | Gemini-2.5 | | GPT-4o | | GPT-4o-mini | | Qwen3-30b | | Qwen3-235b | | Llama-4-m | | Llama-4-s | |
| --- | --- | --- | --- | --- | --- | --- | --- | --- | --- | --- | --- | --- | --- | --- | --- |
| | | $\theta_{ratio}$ | $D_W$ | $\theta_{ratio}$ | $D_W$ | $\theta_{ratio}$ | $D_W$ | $\theta_{ratio}$ | $D_W$ | $\theta_{ratio}$ | $D_W$ | $\theta_{ratio}$ | $D_W$ | $\theta_{ratio}$ | $D_W$ |
| **SummEval** | *Relevance* | **0.96** | 0.30 | 1.55 | 0.30 | 1.39 | 0.34 | 1.30 | 0.42 | 1.47 | 0.21 | **1.07** | 0.25 | 1.30 | 0.28 |
| | *Consistency* | 1.36 | 0.26 | 1.89 | 0.55 | 2.20 | 0.57 | 1.57 | 0.34 | 1.36 | 0.42 | 1.29 | 0.41 | 1.65 | 0.55 |
| | *Fluency* | 1.71 | 0.48 | 1.63 | 0.42 | 2.29 | 0.57 | 1.97 | 0.49 | 1.76 | 0.50 | 1.33 | 0.47 | 1.57 | 0.49 |
| | *Coherence* | 1.24 | 0.26 | 1.50 | 0.27 | 1.66 | 0.31 | 1.53 | 0.30 | 1.55 | 0.32 | **0.98** | 0.24 | 1.30 | 0.27 |
| **TopicalChat** | *Understandability* | 2.46 | 0.33 | 2.59 | 0.33 | 2.59 | 0.34 | 2.56 | 0.36 | 2.89 | 0.36 | 2.29 | 0.31 | 2.75 | 0.33 |
| | *Naturalness* | 1.71 | 0.48 | 3.19 | 0.54 | 2.52 | 0.55 | 2.12 | 0.41 | 1.96 | 0.50 | 2.31 | 0.50 | 2.35 | 0.41 |
| | *Coherence* | 1.79 | 0.45 | 2.41 | 0.55 | 2.01 | 0.43 | 1.93 | 0.43 | 1.97 | 0.53 | 1.88 | 0.43 | 1.97 | 0.43 |
| | *Engagingness* | 2.21 | 0.42 | 3.07 | 0.44 | 2.16 | 0.43 | 2.08 | 0.51 | 2.21 | 0.54 | 2.05 | 0.44 | 2.75 | 0.51 |
| | *Groundedness* | 2.51 | 0.56 | 2.38 | 0.56 | 2.41 | 0.57 | 2.46 | 0.54 | 2.38 | 0.56 | 2.47 | 0.57 | 2.46 | 0.55 |
| **HelpSteer-2** | *Helpfulness* | 1.80 | 0.33 | 1.83 | 0.33 | 1.76 | 0.32 | 1.61 | 0.31 | 1.58 | 0.33 | 1.86 | 0.30 | 1.65 | 0.31 |
| | *Correctness* | 1.46 | 0.28 | 1.55 | 0.29 | 1.77 | 0.32 | 1.52 | 0.29 | 1.60 | 0.30 | 1.76 | 0.34 | 1.75 | 0.34 |
| | *Coherence* | **1.10** | 0.25 | 1.29 | 0.23 | 1.34 | 0.15 | 1.03 | 0.16 | 1.48 | 0.15 | 1.24 | 0.29 | 1.20 | 0.21 |
| | *Complexity* | 1.26 | 0.34 | 1.16 | 0.30 | 1.50 | 0.32 | **1.10** | 0.46 | 1.30 | 0.36 | 1.32 | 0.28 | 1.40 | 0.30 |
| | *Verbosity* | **1.02** | 0.37 | 1.21 | 0.45 | 1.46 | 0.17 | 0.99 | 0.53 | 1.18 | 0.52 | 1.40 | 0.22 | 1.33 | 0.23 |
| **VIEScore** | *CIG-SC* | 2.08 | 0.45 | 1.92 | 0.47 | 1.48 | 0.45 | 1.74 | 0.46 | 1.37 | 0.57 | 1.48 | 0.43 | 0.86 | 0.38 |
| | *PQ* | 3.35 | 0.51 | 3.03 | 0.50 | 3.11 | 0.43 | 3.09 | 0.46 | 2.61 | 0.56 | 2.76 | 0.42 | 2.03 | 0.33 |
| | *MIE-SC* | 1.87 | 0.55 | 2.42 | 0.43 | 1.96 | 0.55 | 2.15 | 0.49 | 1.79 | 0.56 | 2.00 | 0.56 | 1.60 | 0.52 |
| | *PQ* | 1.56 | 0.49 | 1.98 | 0.39 | 1.84 | 0.43 | 1.75 | 0.43 | 1.73 | 0.43 | 1.28 | 0.45 | 1.12 | 0.47 |
| | *TIE-SC* | 1.99 | 0.51 | 2.06 | 0.51 | 2.29 | 0.45 | 2.20 | 0.47 | 2.09 | 0.50 | 1.85 | 0.52 | 1.71 | 0.51 |
| | *PQ* | 3.86 | 0.61 | 4.40 | 0.60 | 4.14 | 0.52 | 3.75 | 0.60 | 3.92 | 0.58 | 3.47 | 0.53 | 3.91 | 0.58 |

tion specification might affect consistency $C_V$. Providing detailed prompts substantially reduces $C_V$ across criteria, and adding chain-of-thought prompting yields further improvement (e.g., Naturalness: GPT-4o achieves $C_V = 0.01$, Qwen3-30b and Llama-4-m achieve $C_V = 0.06$). However, $\rho$ shows only marginal gains, suggesting that detailed instructions stabilize prompt sensitivity but do not fundamentally improve discriminative capacity. Second, rating scales might affect marginal reliability $\rho$. The 5-point scale shows modest improvement in $\rho$ for graded criteria (Naturalness: 0.91–0.95, Coherence: 0.90–0.95, Engagingness: 0.91–0.94), but the 7-point scale does not yield consistent further gains and occasionally decreases reliability.

These results emphasize that our framework is suitable to capture subtle differences due to prompt engineering and scale adjustment. Though prior work reported similar observations that more explicit instructions or chain-of-thought prompting can lead to increased evaluation stability (Wang et al., 2024b; Wagner et al., 2024; Saha et al., 2025), our result further implies guideline for designing a good judgment criteria: More detailed instructions tend to stabilize prompt consistency ($C_V$), while scale adjustment affects measurement reliability ($\rho$).

## 6. Results 2: Human Alignment

Having established intrinsic consistency in Phase 1, we now examine how LLM judges' quality perception compares to human judgment. Table 2 presents the Phase 2 diagnostic results with two metrics. (1) $\theta_{ratio}$ (discrimination breadth ratio) measures whether LLMs perceive quality differences

with similar breadth to humans: $\theta_{ratio} < 1$, $\theta_{ratio} \approx 1$ and $\theta_{ratio} > 1$ indicate hypersensitive, near-human, insensitive calibration. (2) $D_W$ (Wasserstein distance) measures distributional divergence from human judgments, where lower values indicate closer alignment. Similar to Phase 1, we analyze both metrics along the same three axes.

### 6.1. Discrimination Breadth Ratio ($\theta_{ratio}$)

**Modality: VIEScore exhibits broader discrimination than NLP subtasks.** VIEScore exhibits the highest $\theta_{ratio}$ values, particularly for perceptual quality (PQ) subtasks (2.03–4.40), indicating that VLMs amplify quality differences far beyond human perception. Criteria in NLP benchmarks show more moderate insensitivity (1.0–2.5), with several criteria approaching near-human calibration ($\theta_{ratio} \approx 1$).

**Model: No consistent scaling effect on calibration.** Unlike Phase 1, model scale does not show consistent effects on $\theta_{ratio}$. Qwen3-235B and Qwen3-30B achieve comparable values across benchmarks, as do GPT-4o and GPT-4o-mini. This suggests that discrimination calibration is depending on task rather than model capacity.

**Task: HelpSteer-2 shows most appropriate calibration.** HelpSteer-2 Coherence and Verbosity most frequently achieve $\theta_{ratio} \approx 1$ (e.g., Qwen3-30b: 1.03, 0.99; Gemini-2.5: 1.10, 1.02). SummEval Relevance and Coherence also show near-human calibration for select models. In contrast, TopicalChat and VIEScore PQ consistently exhibit insensitivity across all models, suggesting these tasks inherently elicit amplified quality perception in LLMs.

*Table 3.* Ablation results on TopicalChat with varying instruction detail, chain-of-thought prompting, and rating scales. CV denotes prompt consistency (lower is better); $\rho$ denotes marginal reliability (higher is better). Highlighted cells indicate CV $\leq 0.10$ and $\rho \geq 0.70$. Gemini-2.5 refers to Gemini 2.5 Flash; Llama-4-m and Llama-4-s refer to Llama-4-Maverick and Llama-4-Scout, respectively.

| Topical Chat | | Gemini-2.5 | | GPT-4o | | GPT-4o-mini | | Qwen3-30b | | Qwen3-235b | | Llama-4-m | | Llama-4-s | |
|---|---|---|---|---|---|---|---|---|---|---|---|---|---|---|---|
| | | $C_V$ | $\rho$ | $C_V$ | $\rho$ | $C_V$ | $\rho$ | $C_V$ | $\rho$ | $C_V$ | $\rho$ | $C_V$ | $\rho$ | $C_V$ | $\rho$ |
| **Detail Prompt** | *Understandability* | 0.12 | 0.36 | 0.21 | 0.43 | 0.11 | 0.56 | 0.12 | 0.67 | 0.14 | 0.40 | 0.13 | 0.47 | 0.43 | 0.36 |
| | *Naturalness* | 0.12 | 0.84 | **0.10** | **0.83** | 0.21 | 0.64 | 0.14 | 0.88 | 0.13 | 0.87 | 0.19 | 0.84 | 0.44 | 0.72 |
| | *Coherence* | 0.20 | 0.78 | **0.10** | **0.78** | 0.21 | 0.84 | 0.13 | 0.83 | 0.21 | 0.81 | 0.14 | 0.84 | 0.25 | 0.81 |
| | *Engagingness* | 0.11 | 0.76 | 0.16 | 0.69 | 0.17 | 0.70 | 0.13 | 0.80 | 0.23 | 0.80 | **0.10** | **0.82** | 0.14 | 0.61 |
| | *Groundedness* | **0.04** | **0.73** | 0.03 | 0.68 | 0.07 | 0.71 | **0.10** | **0.73** | 0.22 | 0.69 | 0.22 | 0.72 | 0.14 | 0.71 |
| **Detail Prompt + CoT** | *Understandability* | 0.13 | 0.50 | 0.22 | 0.58 | 0.28 | 0.60 | 0.14 | 0.64 | 0.44 | 0.46 | 0.14 | 0.67 | 0.18 | 0.66 |
| | *Naturalness* | 0.18 | 0.84 | **0.01** | **0.85** | 0.47 | 0.70 | 0.23 | 0.87 | **0.06** | **0.87** | **0.06** | **0.87** | **0.09** | **0.80** |
| | *Coherence* | 0.21 | 0.82 | **0.10** | **0.79** | 0.37 | 0.81 | 0.18 | 0.82 | **0.10** | **0.84** | **0.06** | **0.82** | 0.53 | 0.79 |
| | *Engagingness* | 0.12 | 0.81 | **0.06** | **0.73** | 0.39 | 0.75 | 0.16 | 0.79 | **0.10** | **0.85** | **0.03** | **0.84** | 0.25 | 0.67 |
| | *Groundedness* | **0.06** | **0.71** | 0.14 | 0.49 | 0.16 | 0.71 | 0.11 | 0.68 | 0.10 | 0.58 | 0.03 | 0.62 | 0.17 | 0.59 |
| **Likert 5** | *Naturalness* | 0.20 | 0.93 | 0.15 | 0.95 | 0.32 | 0.91 | 0.12 | 0.92 | 0.49 | 0.94 | 0.17 | 0.94 | 0.47 | 0.92 |
| | *Coherence* | 0.20 | 0.90 | **0.09** | **0.91** | 0.13 | 0.95 | 0.12 | 0.93 | 0.28 | 0.93 | 0.14 | 0.92 | 0.13 | 0.92 |
| | *Engagingness* | 0.13 | 0.92 | 0.19 | 0.91 | **0.10** | **0.93** | 0.16 | 0.94 | 0.25 | 0.94 | 0.27 | 0.94 | **0.07** | **0.91** |
| **Likert 7** | *Naturalness* | **0.08** | **0.95** | 0.15 | 0.95 | 0.32 | 0.87 | 0.22 | 0.90 | 0.34 | 0.93 | 0.26 | 0.81 | 0.24 | 0.90 |
| | *Coherence* | 0.49 | 0.87 | 0.31 | 0.95 | 0.17 | 0.97 | 0.31 | 0.96 | 0.19 | 0.97 | 0.22 | 0.95 | 0.14 | 0.94 |
| | *Engagingness* | 0.34 | 0.93 | 0.18 | 0.96 | 0.28 | 0.94 | 0.25 | 0.96 | 0.26 | 0.96 | 0.30 | 0.94 | 0.26 | 0.92 |
| **Detail Prompt + CoT + Likert 5** | *Naturalness* | 0.25 | 0.93 | **0.06** | **0.94** | 0.29 | 0.88 | 0.16 | 0.93 | 0.22 | 0.95 | 0.11 | 0.92 | 0.55 | 0.92 |
| | *Coherence* | 0.18 | 0.89 | 0.18 | 0.89 | **0.10** | **0.94** | **0.09** | **0.91** | 0.19 | 0.92 | 0.17 | 0.90 | 0.33 | 0.89 |
| | *Engagingness* | 0.15 | 0.90 | **0.08** | **0.87** | 0.26 | 0.92 | **0.07** | **0.90** | 0.11 | 0.94 | 0.17 | 0.94 | 0.27 | 0.88 |

## 6.2. Distributional Alignment ($D_W$)

**Modality: No systematic modality effect.** $D_W$ shows no clear difference between NLP and vision benchmarks. Both modalities exhibit similar ranges (0.15–0.61), with within-modality difference exceeding between-modality difference.

**Model: Benchmark effects dominate model effects.** All models exhibit similar $D_W$ patterns within each benchmark, with minimal variation across model families or scales. This suggests that distributional alignment is primarily determined by task characteristics.

**Task: HelpSteer-2 Coherence achieves closest alignment.** HelpSteer-2 Coherence achieves the lowest $D_W$ across all benchmarks (0.15–0.29), indicating closest distributional alignment with human judgments. SummEval Coherence also performs well (0.24–0.32). VIEScore TIE-PQ shows the highest $D_W$ (0.51–0.61), consistent with its extremely high $\theta_{\text{ratio}}$ values.

## 6.3. Discussion

Phase 2 analysis reveals that $\theta_{\text{ratio}}$ and $D_W$ capture complementary but incomplete aspects of human alignment. $\theta_{\text{ratio}}$ identifies systematic insensitivity ($\theta_{\text{ratio}} > 1$) across nearly all model-task combinations, indicating that LLM judges consistently perceive quality differences more broadly than humans. $D_W$ quantifies overall distributional divergence but shows no clear effect of modality.

However, these aggregate metrics mask a critical distinction that emerges only through score-level analysis. Figure 1

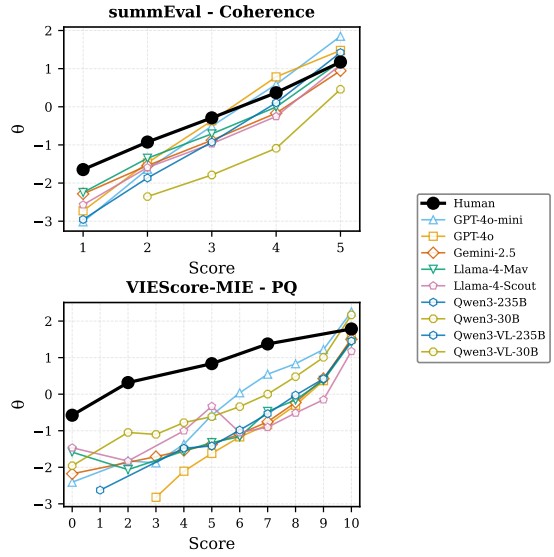

*Figure 1.* Median latent quality ($\theta$) by human score for SummEval-Coherence (up) and VIEScore-MIE PQ (down). Each line represents a different judge model; the black line denotes human.

visualizes the relationship between human scores and estimated $\theta$ for representative NLP and vision tasks. In NLP tasks (SummEval-Coherence), all models exhibit monotonically increasing $\theta$ trajectories parallel to humans. The slopes differ, but the direction is preserved. This pattern indicates a calibration mismatch: LLMs and humans perceive the same quality construct with different scales. In contrast, vision tasks (VIEScore-MIE PQ) reveal non-monotonic patterns where model $\theta$ values decrease or plateau. This suggests a

validity gap[4] between VLMs and humans: VLMs may evaluate different aspects of quality than what humans evaluate.

This interpretation is evidenced by sample-wise Pearson correlations between human rating scores and machine rating scores (Pearson (1896); Appendix F.2): NLP tasks achieve moderate correlations ($r = 0.2$–$0.5$), while vision tasks approach zero. Together, these findings suggest that $\theta_{\text{ratio}} > 1$ carries different implications depending on the underlying pattern. In NLP tasks, the calibration mismatch indicates that LLMs measure the same construct as humans but with different scales. In vision tasks, the calibration mismatch indicates the validity gap. We provide score-level visualizations for all benchmarks in Appendix E.

### 6.4. Practical Recommendations

Our experiments suggest several tendencies that may help practitioners interpret and act on the diagnostic signals, based on the settings considered in this study. When prompt sensitivity (CV) is high, adding more detailed evaluation instructions with explicit rubric definitions tends to improve stability, with additional gains from chain-of-thought prompting (Table 3). When marginal reliability ($\rho$) is low or unstable, adjusting the rating scale may help improve measurement reliability, as different scales yield substantially different $\rho$ values (Table 3). For human alignment, NLP tasks often exhibit patterns consistent with calibration mismatch, suggesting that differences may be partially addressed through post-hoc rescaling. In VIEScore tasks, judges often exhibit patterns indicative of a construct validity gap, suggesting that they may capture different aspects of quality than humans (Figure 1). While our framework identified this tendency, extending the analysis to additional benchmarks remains an important direction for strengthening the generalizability of diagnostic findings.

### 7. Conclusion

We presented a framework for diagnosing reliability of LLM-as-a-Judge by adapting Item Response Theory to the evaluation setting, where prompt variations serve as measurement items and latent quality $\theta$ captures perceived sample quality. From this formulation, we derive four interpretable metrics organized into two sequential phases. Phase 1 metrics ($C_V$ and $\rho$) identify whether inconsistency stems from prompt sensitivity or inadequate use of the rating scale, guiding practitioners toward appropriate adjustments. Phase 2 metrics ($\theta_{\text{ratio}}$ and $D_W$) reveal how LLM judges differ from humans, distinguishing calibration mismatch from validity gap. This framework provides practitioners

with a principled tool for diagnosing LLM judge reliability.

### 8. Limitations

While our framework offers a principled basis for diagnosing LLM-as-a-Judge reliability, four limitations remain. Our framework currently applies only to point-scale judgments; extension to pairwise or open-ended evaluations remains future work. Phase 1 evaluates stability under surface-level perturbations as a minimal prerequisite, while structural reformulations such as full-sentence paraphrasing or rubric reordering are treated as new measurement instruments that should undergo Phase 1 independently. Vision-language evaluation is examined through VIEScore alone, so generalizing the observed tendencies to the entire vision modality requires validation across additional multimodal benchmarks. The GRM models latent quality and prompt effects but does not explicitly capture other reliability concerns such as self-preference, position, and verbosity biases, which can be incorporated via Differential Item Functioning (DIF) in future work. Finally, we diagnose measurement behavior from observed ratings without assessing the faithfulness of judges' internal reasoning, and our experiments focus on English text and a single vision modality, leaving multilingual evaluation and additional modalities as future directions.

### Impact Statement

This work aims to promote more responsible use of LLM-as-a-Judge by offering a diagnostic framework that clarifies when automated evaluators behave as reliable measurement instruments and when they do not. By separating intrinsic consistency from human alignment, the proposed approach helps mitigate the risk of over-trusting LLM-based evaluation in benchmarking and model development. At the same time, our study has several limitations that constrain its broader impact. The framework is currently applicable only to point-scale (rating-based) judgments and does not directly extend to pairwise or open-ended evaluation settings. While we analyze consistency and alignment, we do not assess whether the internal reasoning or explanations produced by LLM judges are faithful to their actual decision processes. In addition, our experiments focus on a limited set of languages and modalities; multilingual evaluation and a wider range of modalities remain unexplored and may exhibit different reliability characteristics. Addressing these limitations is an important direction for future work to ensure that reliability diagnostics for LLM-as-a-Judge generalize across settings and do not introduce unintended biases or misuse.

---

[4]We use 'validity gap' in the sense of construct validity (Cronbach & Meehl, 1955; Salaudeen et al., 2025), referring to a mismatch between the latent construct measured by the LLM judge and the construct intended by human raters.

## Acknowledgments

This research was supported by Basic Science Research Program through the National Research Foundation of Korea(NRF) funded by the Ministry of Education (RS-2025-25434151) and the Institute of Information & Communications Technology Planning & Evaluation (IITP) grant funded by the Korea government (MSIT) [RS-2021-II211341, Artificial Intelligence Graduate School Program (Chung-Ang University)].

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

## The Use of Large Language Models

We used AI assistance tools during the writing process of this manuscript. Specifically, we employed Grammarly for grammar checking, and GPT-5 for language polishing and improving clarity of expression. These tools were used for editorial purposes.

## A. Details of Graded Response Model for LLM Judges

### A.1. Derivation of the Graded Response Model

The main text introduces the Graded Response Model directly. For a $K$-point scale, it specifies the probability that prompt variant $p \in \mathcal{P}$ rates subject $j \in \mathcal{D}$ at score at least $k$ as

$$P(Y_{pj} \geq k \mid \theta_j) = \frac{1}{1 + \exp(-\alpha_p(\theta_j - \beta_{pk}))} \tag{1 restated}$$

Here we re-derive this expression from a simple generative story, so that each parameter has a transparent meaning and the reader can see *why* $\theta_j$ is the right quantity to compare across judges.

**Setup: an exam analogy.** It helps to keep the educational-testing analogy in mind. In IRT, a student of ability $\theta$ answers test items, and each item has its own difficulty and how sharply it separates strong from weak students. We replace the student's ability with the *latent quality* $\theta_j$ of an evaluation subject $j$, and we treat each prompt variant $p$ (original, typo, newline, paraphrase) as an "item" with its own characteristics. A reliable judge is then one whose "items" measure the same underlying quality consistently.

**Step 1: a continuous latent response.** We assume that, before producing a discrete rating, prompt variant $p$ forms an internal continuous impression $Z_{pj}$ of subject $j$:

$$Z_{pj} = \theta_j + \varepsilon_{pj}, \qquad \varepsilon_{pj} \sim \text{Logistic}(0,\, 1/\alpha_p) \tag{8}$$

The impression is centered at the subject's true quality $\theta_j$, plus measurement noise $\varepsilon_{pj}$. The noise scale $1/\alpha_p$ controls how cleanly the variant separates good subjects from bad ones. When $\alpha_p$ is large the noise variance $\pi^2/(3\alpha_p^2)$ is small, so $Z_{pj}$ stays close to $\theta_j$ and the variant discriminates sharply. This is exactly why $\alpha_p$ is called the *discrimination* parameter. (We use a logistic noise distribution rather than a Gaussian simply because it yields the clean closed form below; the two are nearly indistinguishable in practice.)

**Step 2: turning the impression into a discrete rating.** A judge does not report $Z_{pj}$ itself; it outputs an integer score by deciding which band the impression falls into. We model this with the increasing threshold sequence $\{\beta_{pk}\}_{k=1}^{K}$ of Eq. (1), where $\beta_{pk}$ is the quality level at which the judge becomes willing to assign score $k$ or higher. Adopting the boundary convention $\beta_{p1} = -\infty$ (every subject earns at least the lowest score) and $\beta_{p,K+1} = +\infty$, the rating is

$$Y_{pj} = k \iff \beta_{pk} \leq Z_{pj} < \beta_{p,k+1} \tag{9}$$

**Step 3: the probability of scoring at least $k$.** The event "the rating is at least $k$" is the same as "the impression cleared the $k$-th cut point":

$$P(Y_{pj} \geq k \mid \theta_j) = P(Z_{pj} \geq \beta_{pk}) = P(\varepsilon_{pj} \geq \beta_{pk} - \theta_j) \tag{10}$$

The last step substitutes $Z_{pj} = \theta_j + \varepsilon_{pj}$ and moves $\theta_j$ to the other side. So everything reduces to asking how often the noise exceeds the gap between the cut point $\beta_{pk}$ and the true quality $\theta_j$.

**Step 4: plugging in the logistic noise.** The logistic noise has the convenient CDF $F(t) = [1 + \exp(-\alpha_p t)]^{-1}$, so its survival function (the probability of exceeding $t$) is

$$P(\varepsilon_{pj} \geq t) = 1 - F(t) = \frac{1}{1 + \exp(\alpha_p t)} \tag{11}$$

Setting $t = \beta_{pk} - \theta_j$ gives exactly the GRM of Eq. (1):

$$P(Y_{pj} \geq k \mid \theta_j) = \frac{1}{1 + \exp\big( - \alpha_p(\theta_j - \beta_{pk})\big)} \tag{12}$$

Reading the formula back off this story: the probability of a high rating grows as the true quality $\theta_j$ rises above the cut point $\beta_{pk}$, and the steepness of that growth is set by $\alpha_p$. The ordered constraint $\beta_{p,k} < \beta_{p,k+1}$ ensures these cumulative probabilities decrease monotonically as $k$ increases, as they must.

**Step 5: from cumulative to exact ratings.** The probability of *exactly* score $k$ is the difference of two adjacent cumulative probabilities,

$$P(Y_{pj} = k \mid \theta_j) = P(Y_{pj} \geq k \mid \theta_j) - P(Y_{pj} \geq k + 1 \mid \theta_j) \tag{13}$$

which is guaranteed non-negative precisely because the thresholds are ordered. When $K = 2$ only one threshold survives and the model collapses to the two-parameter logistic (2PL) form, which is what we use for the binary criteria.

**Why $\theta_j$ is comparable across judges (the key point for RQ1).** A natural worry is whether $\theta_j$ is even well-defined: the latent scale could be stretched or shifted arbitrarily. Concretely, the model is unchanged if we rescale $\theta \mapsto a\theta + c$, $\beta \mapsto a\beta + c$, $\alpha \mapsto \alpha/a$ for any $a > 0$. We remove this freedom by anchoring the scale with the prior $\theta_j \sim \mathcal{N}(0,1)$, which fixes the location and spread of the latent dimension once and for all; the priors $\alpha_p \sim \text{LogNormal}(0, 0.5)$ and $\beta_{pk} \sim \mathcal{N}(0,1)$ are only weakly informative.

The more important point is *where the information about $\theta_j$ comes from*. Every prompt variant of a given judge shares the *same* $\theta_j$ for a subject, while each variant keeps its own $(\alpha_p, \boldsymbol{\beta}_p)$. So the $|\mathcal{P}|$ variants act like repeated, independently-distorted measurements of one quantity, and they jointly pin it down. Counting parameters, we fit $|\mathcal{D}| + |\mathcal{P}| + |\mathcal{P}|(K - 1)$ unknowns against $|\mathcal{P}| \cdot |\mathcal{D}|$ observed ratings, so $\theta_j$ is over-determined as soon as $|\mathcal{P}| \geq 2$ (given $|\mathcal{D}| \gg K$). Because $\theta_j$ is thus identified by the subject's behavior across variants—rather than by any single prompt or by the raw score distribution—it reflects an intrinsic property of subject $j$ instead of the quirks of one measurement instrument. This is exactly the property that lets us compare judges that use the rating scale very differently, and it is the foundation for the $\theta$-based analysis in Section 3.

## A.2. Model Fitting

We implement the GRM using PyMC 5.25.1 (Salvatier et al., 2016) with the NumPyro (Phan et al., 2019) backend, and use ArviZ 0.23.0 for posterior diagnostics. Because the GRM has no closed-form posterior under our priors, we rely on Markov chain Monte Carlo rather than point estimation, which lets us propagate estimation uncertainty into the downstream metrics (e.g., the marginal reliability $\rho$, which explicitly depends on the posterior variance of $\theta_j$).

For posterior inference we use the NUTS, a self-tuning variant of Hamiltonian Monte Carlo that is well suited to the continuous, moderately correlated parameters $(\theta, \alpha, \beta)$ in our model. We run 4 independent chains with 1000 warmup iterations and 1000 sampling iterations each, yielding 4000 posterior draws per parameter. Running multiple chains from different initializations lets us assess convergence by comparing within- and between-chain behavior, and the 1000 warmup steps allow NUTS to adapt its step size and mass matrix before sampling begins. We set the target acceptance rate to 0.95, higher than the default 0.8; this produces smaller, more careful steps that reduce divergent transitions, which is helpful given the ordered constraint on the thresholds $\beta_{pk}$. We use `adapt_diag` initialization, which starts from a diagonal estimate of the mass matrix, and fix the random seed to 42 so that the full sampling procedure is reproducible. We apply the following priors: $\theta_j \sim \mathcal{N}(0,1)$ for latent quality, $\alpha_p \sim \text{LogNormal}(0, 0.5)$ for discrimination parameters, and $\beta_{pk} \sim \mathcal{N}(0,1)$ with an ordered constraint ($\beta_{pk} < \beta_{p,k+1}$ for all $k < K$) for threshold parameters. For binary responses (K=2), we fit a 2-parameter logistic model with the same priors for $\theta$ and $\alpha$, and $\beta_p \sim \mathcal{N}(0,1)$ without ordering.

For convergence diagnostics, we compute $\hat{R}$ and effective sample size (ESS) using ArviZ. We also compute leave-one-out cross-validation (LOO) with Pareto-$k$ diagnostics to identify influential observations.

## A.3. Implementation Details

All experiments are conducted with Python 3.10.19 on an AMD EPYC 7313 16-Core Processor.

# B. Details of Prompt Variation

All prompt variations are generated with a fixed random seed (42). Once generated, variations are saved and applied identically to all models within each dataset.

## B.1. Typo

We first compute attention weights from the final layer of Qwen3-8B for all tokens in the prompt. After excluding placeholders (e.g., `{context}`) and protected keywords (metric names, "json", "format", etc.), we select the top-5 high-attention tokens. We then apply character-level perturbations to these tokens using the AugLy library with `aug_word_p=1.0` and `aug_char_p=0.5`. When the library fails to generate a valid typo, we swap two adjacent characters as a fallback.

## B.2. Newline

We split the prompt into segments by newline characters, then randomly select up to three positions to insert additional newlines. Placeholders containing evaluation samples remain unchanged.

## B.3. Paraphrase

We extract verbs (VB, VBD, VBG, VBN, VBP, VBZ) and adjectives (JJ) using NLTK POS tagger. After excluding protected keywords, we prompt GPT-4o-mini to generate synonyms for five tokens. The model is instructed to replace only the selected tokens while preserving the original part-of-speech and meaning.

## B.4. Implementation Details

We use Python 3.10, AugLy 1.0.0, NLTK 3.9.2, and Transformers 4.51.3. Attention extraction is performed on a single NVIDIA RTX 6000 Ada.

## B.5. Example

Tables 4 and 5 present example prompt variations for the SUMMEVAL and HELPSTEER benchmarks. The variations introduce minimal surface-level perturbations, enabling analysis of judging stability under equivalent evaluation conditions.

*Table 4.* Example prompt variations for the SummEval benchmark.

**SummEval**

```
"[Typo VARIATIONS]": {
    ONLY -> LONLY (attention: 0.0038),
    JSON -> SJON (attention: 0.0020),
    task -> tsak (attention: 0.0018),

}

"[NEWLINE VARIATIONS]": {
    Extra newlines inserted after line 9,
    Extra newlines inserted after line 10,
    Extra newlines inserted after line 14
}

"[PARAPHRASE VARIATIONS]": {
    evaluate -> assess,
    written -> composed,
    follow -> adhere,
    aware -> cognizant,
    proposed -> suggested
}
```

*Table 5.* Example prompt variations for the HelpSteer-2 benchmark.

**HelpSteer-2**

```
"[Typo VARIATIONS]": {
    Your -> Youe (attention: 0.0608),
    Please -> lPease (attention: 0.0039),
    only -> onyl (attention: 0.0038),

}
"[NEWLINE VARIATIONS]": {
    Extra newlines inserted after line 2,
    Extra newlines inserted after line 6,
    Extra newlines inserted after line 8
}

"[PARAPHRASE VARIATIONS]": {
    expert -> specialist,
    assess -> evaluate,
    following -> subsequent,
    consistent -> coherent,
    sophisticated -> complex
}
```

## C. Prompt Template

We adopt prompt templates from the original papers: SummEval and TopicalChat, HelpSteer-2 (Wang et al., 2024c), and VIEScore (Ku et al., 2024a). For some benchmarks, we modify the output instruction to enforce JSON format. The detailed description and chain-of-thought (CoT) variants in our ablation study are our modifications.

### C.1. SummEval

```
In this task you will evaluate the quality of summaries written for a news article.
To correctly solve this task, follow these steps:
1. Carefully read the news article, be aware of the information it contains.
2. Read the proposed summary.
3. Rate the summary on a scale from 1 (worst) to 3 (best) by its relevance,
consistency, fluency, and coherence.

Definitions:
Relevance: The rating measures how well the summary captures the key points of the
article.
Consistency: The rating measures whether the facts in the summary are consistent with
the facts in the original article.
Fluency: This rating measures the quality of individual sentences.
Coherence: This rating measures the quality of all sentences collectively.

--
News Article:
{ARTICLE}

--
Summary:
{SUMMARY}

--
Please provide your ratings in the following JSON format ONLY:
"relevance": <1-5>, "consistency": <1-5>, "fluency": <1-5>, "coherence": <1-5>
```

## C.2. TopicalChat

### C.2.1. ORIGINAL PROMPT

```
In this task, you will evaluate the quality of a dialogue response in a
knowledge-grounded conversation.

Please carefully follow the steps below:
1. Read the dialogue context to understand the conversation history.
2. Read the given response.
3. Rate the summary using integer scores understandability (0-1), naturalness (1-5),
coherence (1-5), engagingness (1-5), groundedness (0-1).

Definitions:
Understandability: judge whether the response is understandable.
Naturalness: Judge whether a response is like something a person would naturally say.
Coherence: Determine whether this response serves as a valid continuation of the
previous conversation.
Engagingness: Determine if the response is interesting or dull.
Groundedness: Given the fact that this response is conditioned on, determine whether
this response uses that fact.
--
Dialogue Context:
{CONTEXT}

--
Response:
{RESPONSE}

--
Please provide your ratings in the following JSON format ONLY:
"understandability": <0-1>, "naturalness": <1-3>, "coherence": <1-3>, "engagingness":
<1-3>, "groundedness": <0-1>
```

## C.2.2. DETAILED PROMPT

```
In this task, you will evaluate the quality of a dialogue response in a
knowledge-grounded conversation.

Please carefully follow the steps below:
1. Read the dialogue context to understand the conversation history.
2. Read the given response.
3. Rate the summary using integer scores understandability (0-1), naturalness (1-5),
coherence (1-5), engagingness (1-5), groundedness (0-1).

Definitions:
Understandability: Judge whether the response is understandable in terms of basic
meaning and intent. A response is understandable if a reader can clearly grasp what is
being said without ambiguity, guesswork, or repeated re-reading. Assign a lower score
if the response contains grammatical errors, broken sentence structure, unclear
references, or logical gaps that make the meaning difficult or impossible to
understand.

Naturalness: Judge whether the response sounds like something a real person would
naturally say in a conversational setting. Consider fluency, phrasing, tone, and
whether the language feels organic rather than robotic, translated, or overly formal.
Lower scores indicate stiff, mechanical, repetitive, or unnatural wording, while
higher scores indicate smooth, human-like conversational language.

Coherence: Determine whether this response serves as a valid and appropriate
continuation of the previous conversation. Consider whether the response directly
addresses the preceding dialogue, stays on topic, and follows logically from what was
said before. Lower scores indicate topic drift, non sequiturs, contradictions, or
failure to respond to the prior turn.

Engagingness: Determine whether the response is interesting, informative, or engaging
for a human reader. Lower scores indicate dull, generic, minimal, or boilerplate
responses that add little value to the conversation. Higher scores indicate responses
that provide useful information, thoughtful elaboration, or otherwise maintain the
reader's interest.

Groundedness: Given the fact or knowledge that this response is conditioned on,
determine whether the response correctly uses and reflects that information. Assign a
lower score if the response introduces unsupported claims, contradicts the provided
context, or ignores relevant grounding facts. Assign a higher score if the response is
consistent with, accurately reflects, and appropriately incorporates the given factual
information.
--
Dialogue Context:
{CONTEXT}

--
Response:
{RESPONSE}

--
Please provide your ratings in the following JSON format ONLY:
"understandability": <0-1>, "naturalness": <1-3>, "coherence": <1-3>, "engagingness":
<1-3>, "groundedness": <0-1>
```

## C.2.3. DETAILED + CoT PROMPT

```
In this task, you will evaluate the quality of a dialogue response in a
knowledge-grounded conversation.

Please carefully follow the steps below:
1. Read the dialogue context to understand the conversation history.
2. Read the given response.
3. Rate the summary using integer scores understandability (0-1), naturalness (1-5),
coherence (1-5), engagingness (1-5), groundedness (0-1).

Definitions:
Understandability: Judge whether the response is understandable in terms of basic
meaning and intent. A response is understandable if a reader can clearly grasp what is
being said without ambiguity, guesswork, or repeated re-reading. Assign a lower score
if the response contains grammatical errors, broken sentence structure, unclear
references, or logical gaps that make the meaning difficult or impossible to
understand.

Naturalness: Judge whether the response sounds like something a real person would
naturally say in a conversational setting. Consider fluency, phrasing, tone, and
whether the language feels organic rather than robotic, translated, or overly formal.
Lower scores indicate stiff, mechanical, repetitive, or unnatural wording, while
higher scores indicate smooth, human-like conversational language.

Coherence: Determine whether this response serves as a valid and appropriate
continuation of the previous conversation. Consider whether the response directly
addresses the preceding dialogue, stays on topic, and follows logically from what was
said before. Lower scores indicate topic drift, non sequiturs, contradictions, or
failure to respond to the prior turn.

Engagingness: Determine whether the response is interesting, informative, or engaging
for a human reader. Lower scores indicate dull, generic, minimal, or boilerplate
responses that add little value to the conversation. Higher scores indicate responses
that provide useful information, thoughtful elaboration, or otherwise maintain the
reader's interest.

Groundedness: Given the fact or knowledge that this response is conditioned on,
determine whether the response correctly uses and reflects that information. Assign a
lower score if the response introduces unsupported claims, contradicts the provided
context, or ignores relevant grounding facts. Assign a higher score if the response is
consistent with, accurately reflects, and appropriately incorporates the given factual
information.
--
Dialogue Context:
{CONTEXT}

--
Response:
{RESPONSE}

--
Please evaluate the quality of the judgement based on whether the judgement is
grounded on the responses and carefully follows the rubric.
Provide some reasoning and analysis to support your evaluation. Next, provide an
integer rating in JSON format.

--
Please provide your ratings in the following JSON format:
"understandability": <0-1>, "naturalness": <1-5>, "coherence": <1-5>, "engagingness":
<1-5>, "groundedness": <0-1>
```

## C.3. HelpSteer-2

```
You are an expert evaluator for answers to user queries. Your task is to assess
responses to user queries based on helpfulness, relevance, accuracy, and clarity.
Calculate the following metrics for the response: User Query: {query} Model Response:
{response} Metrics: 1. helpfulness (1-5): How well does the response help the user? 2.
correctness (1-5): Is the information correct? 3. coherence (1-5): Is the response
logically consistent and well-structured? 4. complexity (1-5): How sophisticated is
the response? 5. verbosity (1-5): Is the response appropriately detailed? Instructions:
Assign a score from 1 (poor) to 5 (excellent) for each metric.
--
Please provide your ratings in the following JSON format ONLY:
"helpfulness": <1-5>, "correctness": <1-5>, "coherence": <1-5>, "complexity": <1-5>,
"verbosity": <1-5>
```

## C.4. VIEScore-CIG-SC

```
RULES: Two images will be provided: The first being a processed image (e.g. Canny
edges, openpose, grayscale, etc.) and the second being an AI-generated image using the
first image as guidance. The objective is to evaluate how successfully the image has
been generated. On scale 0 to 10: A score from 0 to 10 will be given based on the
success in following the prompt. (0 indicates that the second image does not follow
the prompt at all. 10 indicates the second image follows the prompt perfectly.) A
second score from 0 to 10 will rate how well the generated image is following the
guidance image. (0 indicates that the second image does not follow the guidance at all.
10 indicates that the second image is following the guidance image.) Put the score in
a list such that output score = [score1, score2], where 'score1' evaluates the prompt
and 'score2' evaluates the guidance. Text Prompt: text
```

## C.5. VIEScore-TIE-SC

```
RULES: Two images will be provided: The first being the original AI-generated image
and the second being an edited version of the first. The objective is to evaluate how
successfully the editing instruction has been executed in the second image. Note that
sometimes the two images might look identical due to the failure of the image edit. On
scale of 0 to 10: A score from 0 to 10 will be given based on the success of the
editing. (0 indicates that the scene in the edited image does not follow the editing
instructions at all. 10 indicates that the scene in the edited image follows the
editing instruction text perfectly.) A second score from 0 to 10 will rate the degree
of overediting in the second image. (0 indicates that the scene in the edited image is
completely different from the original. 10 indicates that the edited image can be
recognized as a minimally edited yet effective version of the original.) Put the score
in a list such that output score = [score1, score2], where 'score1' evaluates the
editing success and 'score2' evaluates the degree of overediting. Editing instruction:
instruction
```

## C.6. VIEScore-PQ

```
RULES: The image is an AI-generated image. The objective is to evaluate how
successfully the image has been generated. On a scale 0 to 10: A score from 0 to 10
will be given based on image naturalness. ( 0 indicates that the scene in the image
does not look natural at all or gives an unnatural feeling such as a wrong sense of
distance, wrong shadow, or wrong lighting. 10 indicates that the image looks natural.
) A second score from 0 to 10 will rate the image artifacts. ( 0 indicates that the
image contains a large portion of distortion, watermarks, scratches, blurred faces,
unusual body parts, or subjects not harmonized. 10 indicates the image has no
artifacts. ) Put the score in a list such that output score = [naturalness, artifacts]
```

## D. Intrinsic Consistency Results on VIEScore

Table 6 presents intrinsic consistency results on VIEScore under detailed prompts and chain-of-thought prompting.

*Table 6.* VIEScore intrinsic result. Gemini-2.5 refers to Gemini 2.5 Flash; Llama-4-m and Llama-4-s refer to Llama-4-Maverick and Llama-4-Scout, respectively. For VIEScore, Qwen3-30B and Qwen3-235B refer to their VL variants.

| Topical Chat | | Gemini-2.5 | | GPT-4o | | GPT-4o-mini | | Qwen3-30b | | Qwen3-235b | | Llama-4-m | | Llama-4-s | |
|---|---|---|---|---|---|---|---|---|---|---|---|---|---|---|---|
| | | $C_V$ | $\rho$ | $C_V$ | $\rho$ | $C_V$ | $\rho$ | $C_V$ | $\rho$ | $C_V$ | $\rho$ | $C_V$ | $\rho$ | $C_V$ | $\rho$ |
| **Detail Prompt + CoT** | CIG-SC | 1.15 | 0.91 | 0.63 | 0.90 | 0.58 | 0.79 | 0.64 | 0.82 | 0.67 | 0.83 | 0.20 | 0.58 | 0.19 | 0.66 |
| | PQ | 1.12 | 0.93 | 0.68 | 0.93 | 0.51 | 0.72 | 0.52 | 0.86 | 0.85 | 0.89 | 0.21 | 0.66 | 0.27 | 0.75 |
| | MIE-SC | 0.25 | 0.93 | 0.29 | 0.95 | 0.68 | 0.95 | 0.34 | 0.86 | 0.77 | 0.85 | 0.33 | 0.83 | 0.45 | 0.82 |
| | PQ | 0.22 | 0.59 | 0.33 | 0.86 | 0.72 | 0.93 | 0.36 | 0.87 | 0.85 | 0.79 | 0.23 | 0.65 | 0.26 | 0.65 |
| | TIE-SC | 0.99 | 0.92 | 0.63 | 0.94 | 0.75 | 0.96 | 0.79 | 0.87 | 0.39 | 0.87 | 0.51 | 0.85 | 0.26 | 0.84 |
| | PQ | 1.52 | 0.93 | 0.73 | 0.89 | 0.84 | 0.94 | 0.62 | 0.80 | 0.70 | 0.88 | 0.17 | 0.64 | 0.42 | 0.73 |

# E. Score-Level Latent Quality Trajectories

Figures 2–7 provide additional examples of score-level latent quality trajectories across different benchmarks.

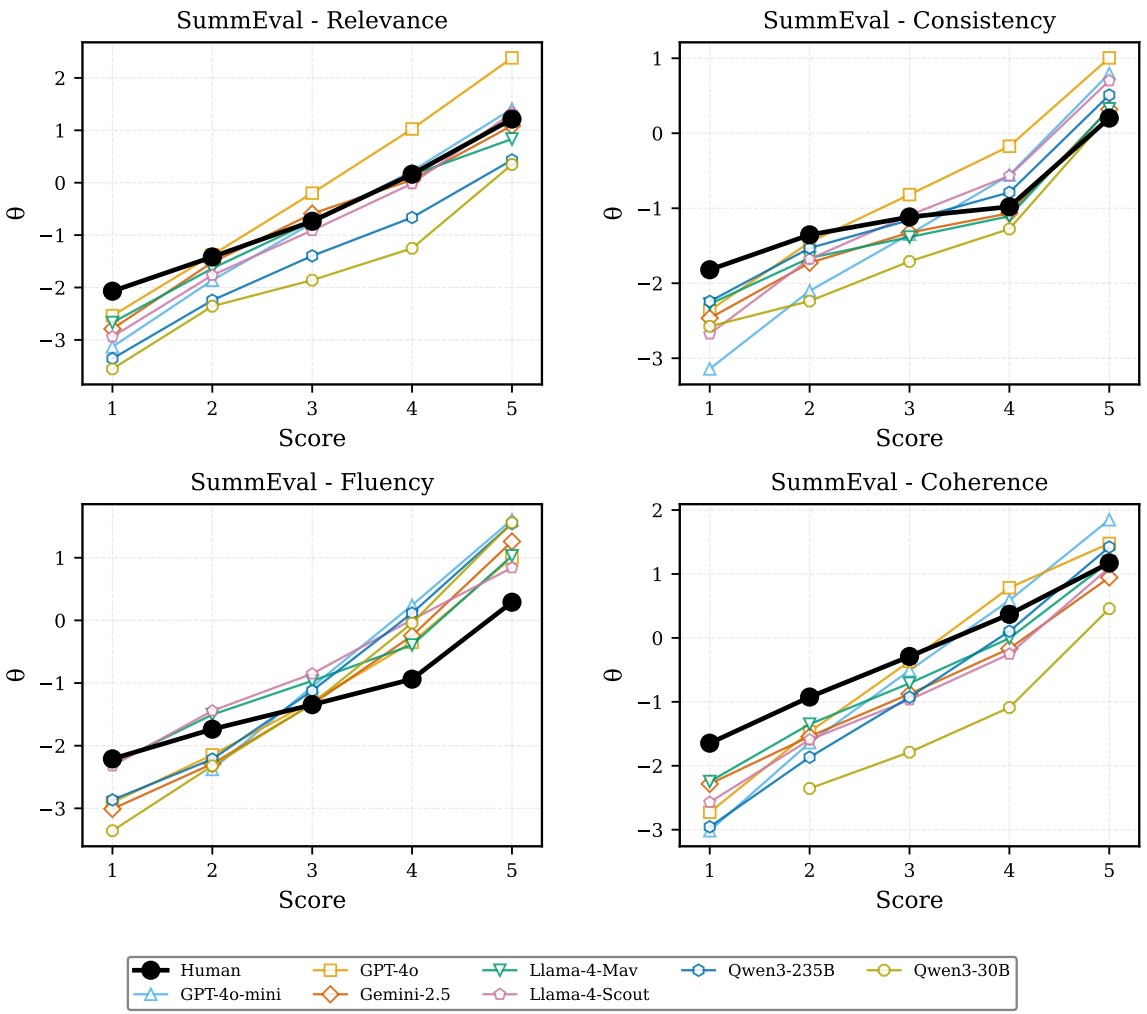

*Figure 2.* Median $\theta$ by human score for all SummEval metrics.

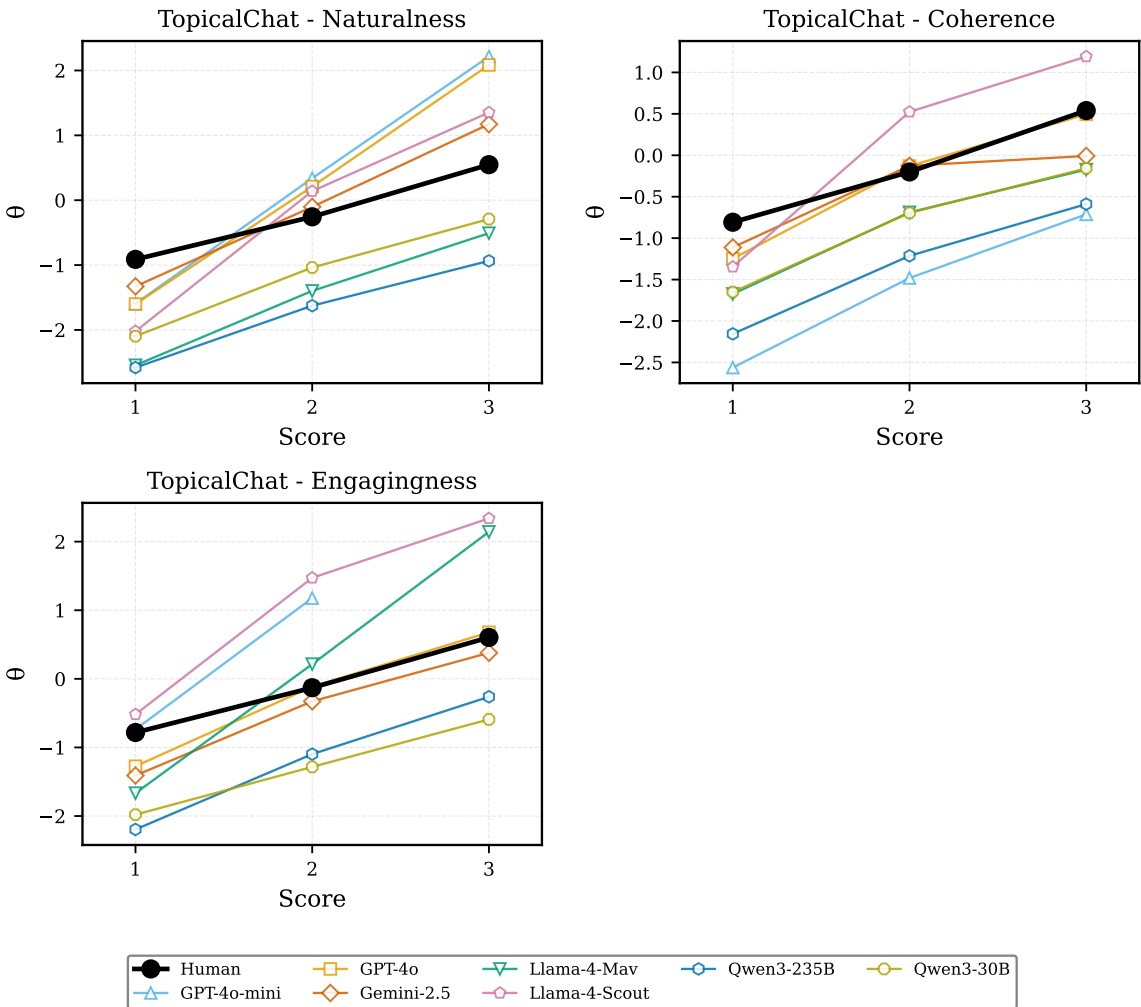

*Figure 3.* Median $\theta$ by human score for all TopicalChat metrics.

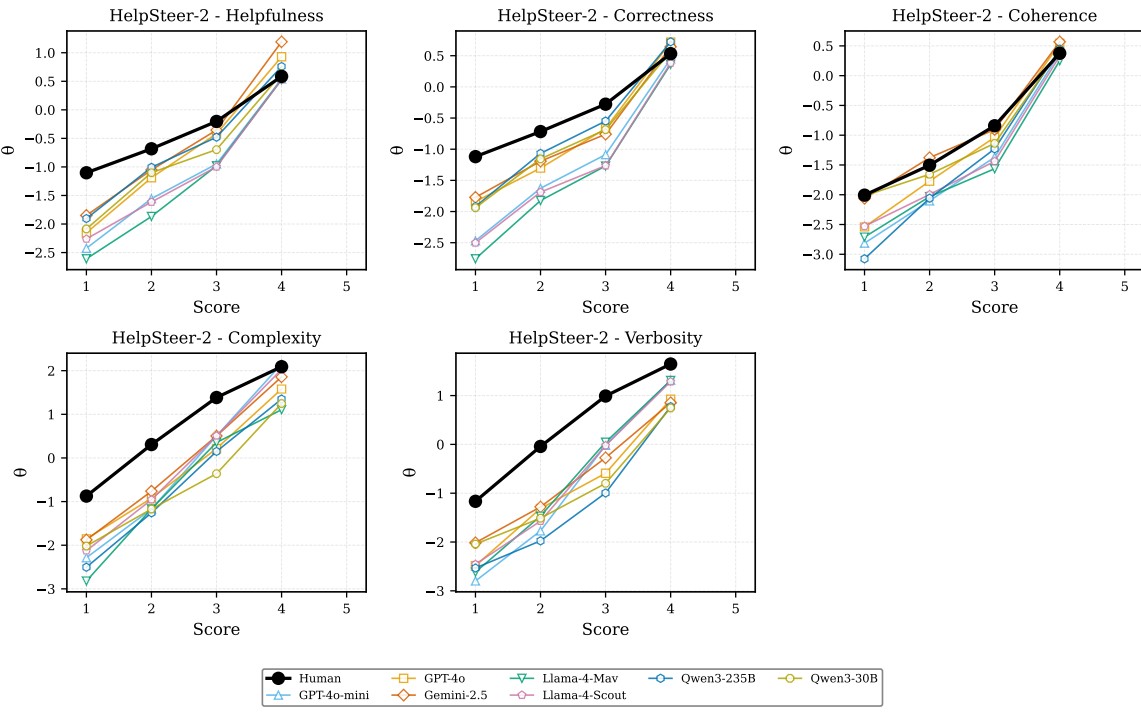

*Figure 4.* Median $\theta$ by human score for all HelpSteer-2 metrics.

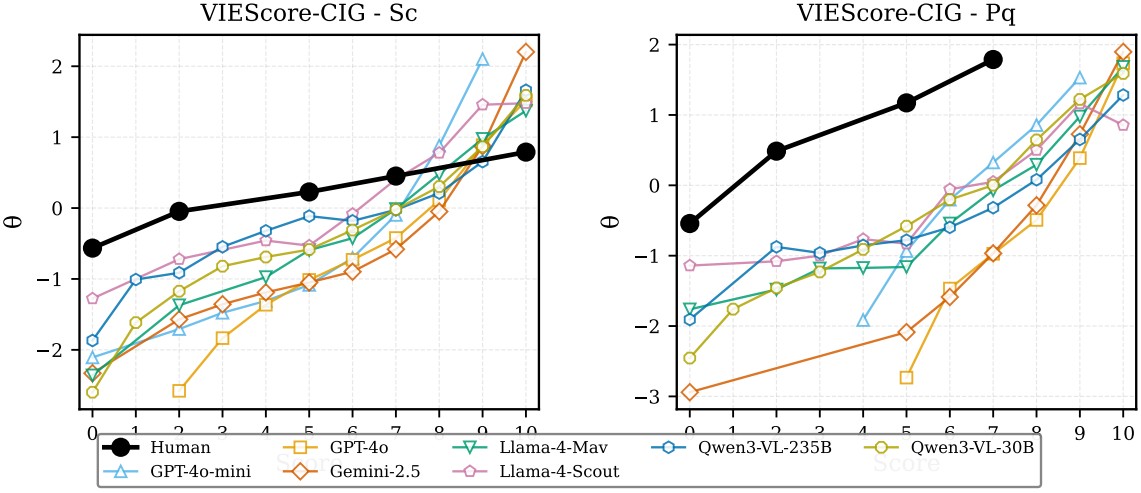

*Figure 5.* Median $\theta$ by human score for VIEScore-CIG (SC and PQ).

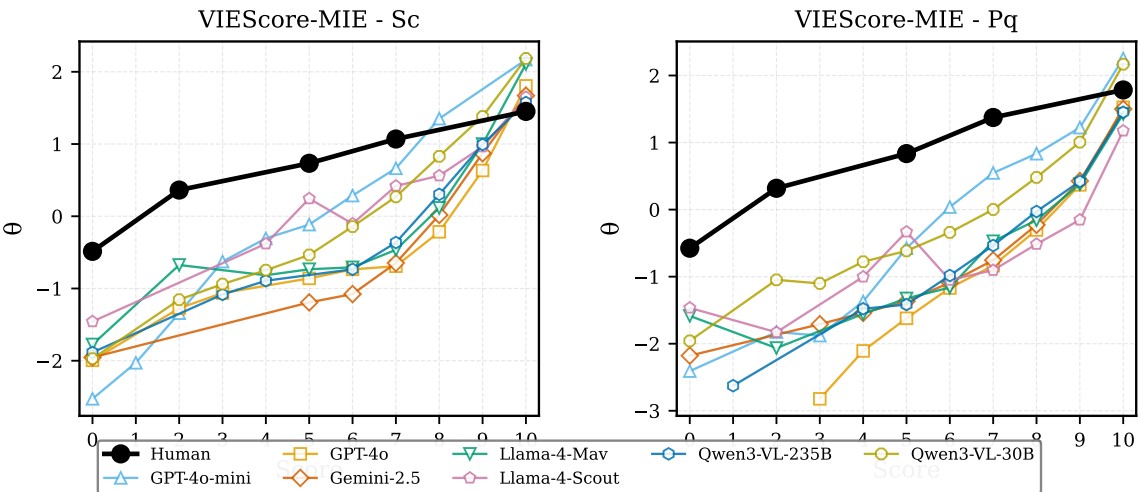

*Figure 6.* Median $\theta$ by human score for VIEScore-MIE (SC and PQ).

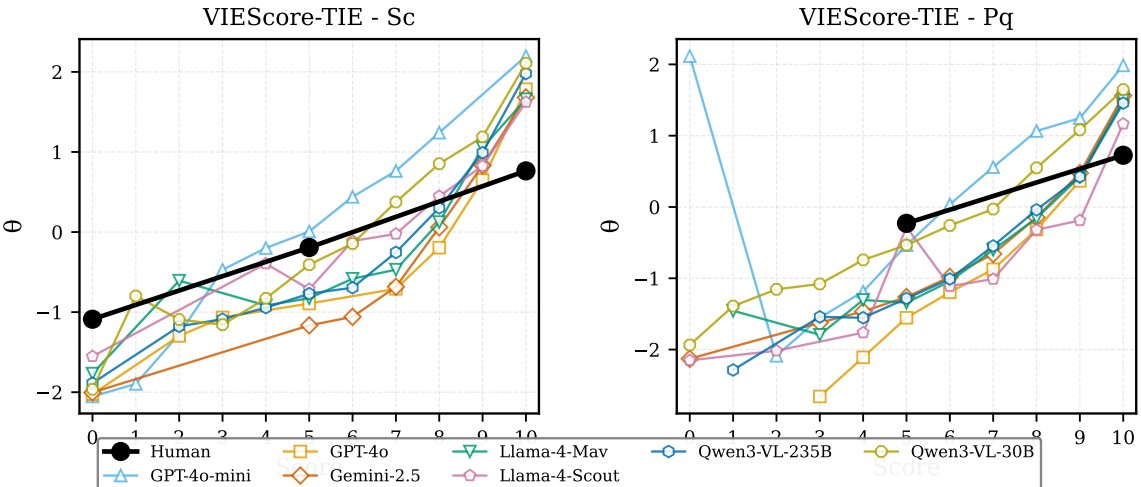

*Figure 7.* Median $\theta$ by human score for VIEScore-TIE (SC and PQ).

## F. Comparison with Existing Reliability Metrics

We compare the proposed diagnostics with existing reliability metrics for both intrinsic consistency and human alignment.

### F.1. Internal Consistency Metrics

Table 7 shows that McDonald's $\omega$ remains consistently high across most settings. In contrast, $C_V$ and $\rho$ exhibit substantially larger variation across benchmarks and models. For example, several VIEScore settings show large differences under prompt variations despite high $\omega$ values, suggesting that $\omega$ does not fully capture variation across prompt conditions.

*Table 7.* McDonald's $\omega$ for internal consistency. Higher values indicate greater internal consistency across prompt variations.

| Benchmark | Criterion | Gemini-2.5 | GPT-4o | GPT-4o-mini | Qwen3-30b | Qwen3-235b | Llama-4-m | Llama-4-s |
|---|---|---|---|---|---|---|---|---|
| SummEval | Relevance | 0.94 | 0.94 | 0.96 | 0.98 | 0.98 | 0.83 | 0.96 |
| | Consistency | 0.99 | 0.96 | 0.96 | 0.99 | 0.99 | 0.93 | 0.98 |
| | Fluency | 0.94 | 0.93 | 0.93 | 0.96 | 0.97 | 0.88 | 0.94 |
| | Coherence | 0.96 | 0.95 | 0.96 | 0.98 | 0.97 | 0.86 | 0.96 |
| TopicalChat | Understandability | 0.90 | 0.97 | 0.97 | 0.96 | 0.98 | 0.96 | 0.91 |
| | Naturalness | 0.96 | 0.96 | 0.94 | 0.96 | 0.97 | 0.95 | 0.85 |
| | Coherence | 0.97 | 0.98 | 0.97 | 0.96 | 0.97 | 0.97 | 0.89 |
| | Engagingness | 0.96 | 0.96 | 0.97 | 0.97 | 0.97 | 0.94 | 0.91 |
| | Groundedness | 0.95 | 0.98 | 0.98 | 0.96 | 0.98 | 0.96 | 0.97 |
| HelpSteer-2 | Helpfulness | 0.92 | 0.97 | 0.97 | 0.97 | 0.98 | 0.91 | 0.95 |
| | Correctness | 0.95 | 0.96 | 0.97 | 0.97 | 0.98 | 0.91 | 0.95 |
| | Coherence | 0.95 | 0.96 | 0.96 | 0.96 | 0.96 | 0.91 | 0.94 |
| | Complexity | 0.88 | 0.92 | 0.93 | 0.94 | 0.95 | 0.81 | 0.94 |
| | Verbosity | 0.90 | 0.94 | 0.89 | 0.94 | 0.93 | 0.82 | 0.88 |
| VIEScore | CIG-SC | 0.95 | 0.99 | 0.96 | 0.94 | 0.97 | 0.92 | 0.80 |
| | CIG-PQ | 0.95 | 0.97 | 0.95 | 0.96 | 0.95 | 0.90 | 0.79 |
| | MIE-SC | 0.97 | 0.97 | 0.95 | 0.95 | 0.97 | 0.96 | 0.92 |
| | MIE-PQ | 0.91 | 0.97 | 0.92 | 0.93 | 0.96 | 0.95 | 0.86 |
| | TIE-SC | 0.97 | 0.99 | 0.95 | 0.95 | 0.98 | 0.96 | 0.92 |
| | TIE-PQ | 0.92 | 0.97 | 0.91 | 0.93 | 0.96 | 0.91 | 0.85 |

### F.2. Human Alignment Metrics

We additionally compare the proposed diagnostics with existing human alignment metrics, including Pearson correlation, Cohen's $\kappa$, and Krippendorff's $\alpha$. Tables 8 and 9 show broadly similar patterns across these metrics, while providing limited insight into the source of disagreement between LLM and human judgments.

*Table 8.* Pearson correlation between human and LLM judgments. Gemini-2.5 refers to Gemini 2.5 Flash; Llama-4-m and Llama-4-s refer to Llama-4-Maverick and Llama-4-Scout, respectively. For VIEScore, Qwen3-30B and Qwen3-235B refer to their VL variants. $^{*}p < 0.05$, $^{**}p < 0.01$, $^{***}p < 0.001$.

| | | Gemini-2.5 | GPT-4o | GPT-4o-mini | Qwen3-30b | Qwen3-235b | Llama-4-m | Llama-4-s |
|---|---|---|---|---|---|---|---|---|
| **SummEval** | *Relevance* | 0.49*** | 0.51*** | 0.53*** | 0.54*** | 0.53*** | 0.40*** | 0.51*** |
| | *Consistency* | 0.75*** | 0.61*** | 0.55*** | 0.72*** | 0.73*** | 0.65*** | 0.67*** |
| | *Fluency* | 0.60*** | 0.56*** | 0.45*** | 0.54*** | 0.47*** | 0.49*** | 0.49*** |
| | *Coherence* | 0.54*** | 0.57*** | 0.52*** | 0.51*** | 0.58*** | 0.48*** | 0.52*** |
| **TopicalChat** | *Understandability* | 0.33*** | 0.38*** | 0.34*** | 0.38*** | 0.30*** | 0.25*** | 0.29*** |
| | *Naturalness* | 0.35*** | 0.49*** | 0.26*** | 0.37*** | 0.46*** | 0.40*** | 0.31*** |
| | *Coherence* | 0.23*** | 0.27*** | 0.30*** | 0.26*** | 0.25*** | 0.21*** | 0.21*** |
| | *Engagingness* | 0.40*** | 0.46*** | 0.39*** | 0.40*** | 0.48*** | 0.31*** | 0.34*** |
| | *Groundedness* | 0.52*** | 0.54*** | 0.53*** | 0.40*** | 0.51*** | 0.42*** | 0.44*** |
| **HelpSteer-2** | *Helpfulness* | 0.46*** | 0.49*** | 0.41*** | 0.41*** | 0.47*** | 0.29*** | 0.41*** |
| | *Correctness* | 0.40*** | 0.48*** | 0.47*** | 0.37*** | 0.41*** | 0.34*** | 0.39*** |
| | *Coherence* | 0.35*** | 0.39*** | 0.35*** | 0.30*** | 0.30*** | 0.24*** | 0.27*** |
| | *Complexity* | 0.20*** | 0.40*** | 0.41*** | 0.21*** | 0.26*** | 0.27*** | 0.36*** |
| | *Verbosity* | 0.21*** | 0.31*** | 0.40*** | 0.18*** | -0.03 | 0.31*** | 0.40*** |
| **VIEScore** | *CIG-SC* | 0.01 | -0.15 | -0.15 | -0.09 | -0.11 | -0.25* | -0.17 |
| | *PQ* | 0.02 | 0.03 | -0.05 | 0.12 | 0.08 | 0.22 | -0.02 |
| | *MIE-SC* | 0.04 | -0.03 | 0.05 | -0.06 | -0.08 | -0.12 | -0.07 |
| | *PQ* | 0.02 | 0.07 | 0.16* | 0.14 | 0.10 | 0.07 | -0.13 |
| | *TIE-SC* | 0.08 | -0.03 | 0.12 | 0.09 | 0.11 | 0.03 | 0.05 |
| | *PQ* | 0.06 | 0.07 | -0.13 | 0.05 | 0.04 | 0.03 | 0.02 |

*Table 9.* Human alignment measured by Cohen's $\kappa$ and Krippendorff's $\alpha$ between LLM judge and human ratings.

| | | Gemini-2.5 | | GPT-4o | | GPT-4o-mini | | Qwen3-30b | | Qwen3-235b | | Llama-4-m | | Llama-4-s | |
|---|---|---|---|---|---|---|---|---|---|---|---|---|---|---|---|
| Benchmark | Criterion | $\kappa$ | $\alpha$ | $\kappa$ | $\alpha$ | $\kappa$ | $\alpha$ | $\kappa$ | $\alpha$ | $\kappa$ | $\alpha$ | $\kappa$ | $\alpha$ | $\kappa$ | $\alpha$ |
| SummEval | Relevance | 0.46 | 0.46 | 0.35 | 0.22 | 0.51 | 0.49 | 0.47 | 0.40 | 0.52 | 0.49 | 0.40 | 0.39 | 0.50 | 0.47 |
| | Consistency | 0.75 | 0.67 | 0.48 | 0.12 | 0.48 | 0.15 | 0.71 | 0.56 | 0.70 | 0.53 | 0.64 | 0.54 | 0.64 | 0.40 |
| | Fluency | 0.48 | 0.20 | 0.34 | -0.05 | 0.17 | -0.33 | 0.36 | 0.02 | 0.30 | -0.04 | 0.41 | 0.27 | 0.30 | -0.05 |
| | Coherence | 0.45 | 0.36 | 0.57 | 0.57 | 0.48 | 0.51 | 0.39 | 0.28 | 0.52 | 0.49 | 0.46 | 0.42 | 0.43 | 0.33 |
| TopicalChat | Understandability | 0.23 | 0.16 | 0.32 | 0.28 | 0.28 | 0.24 | 0.34 | 0.31 | 0.20 | 0.11 | 0.19 | 0.13 | 0.24 | 0.20 |
| | Naturalness | 0.32 | 0.29 | 0.39 | 0.37 | 0.18 | 0.17 | 0.36 | 0.36 | 0.44 | 0.42 | 0.38 | 0.37 | 0.23 | 0.20 |
| | Coherence | 0.18 | 0.07 | 0.20 | 0.07 | 0.26 | 0.21 | 0.26 | 0.25 | 0.22 | 0.17 | 0.21 | 0.22 | 0.15 | 0.04 |
| | Engagingness | 0.24 | 0.04 | 0.29 | 0.17 | 0.28 | 0.25 | 0.37 | 0.34 | 0.42 | 0.37 | 0.26 | 0.22 | 0.16 | -0.11 |
| | Groundedness | 0.48 | 0.47 | 0.51 | 0.50 | 0.52 | 0.51 | 0.39 | 0.39 | 0.50 | 0.49 | 0.42 | 0.41 | 0.41 | 0.39 |
| HelpSteer-2 | Helpfulness | 0.45 | 0.36 | 0.45 | 0.36 | 0.30 | 0.14 | 0.38 | 0.25 | 0.45 | 0.35 | 0.19 | 0.02 | 0.33 | 0.13 |
| | Correctness | 0.39 | 0.27 | 0.44 | 0.32 | 0.36 | 0.17 | 0.35 | 0.23 | 0.41 | 0.32 | 0.22 | 0.00 | 0.30 | 0.07 |
| | Coherence | 0.31 | 0.29 | 0.39 | 0.29 | 0.33 | 0.23 | 0.28 | 0.23 | 0.30 | 0.25 | 0.21 | 0.10 | 0.26 | 0.17 |
| | Complexity | 0.13 | -0.09 | 0.19 | -0.13 | 0.18 | -0.18 | 0.08 | -0.31 | 0.11 | -0.31 | 0.10 | -0.33 | 0.20 | -0.08 |
| | Verbosity | 0.11 | -0.24 | 0.12 | -0.28 | 0.17 | -0.28 | 0.07 | -0.36 | -0.01 | -0.62 | 0.15 | -0.25 | 0.21 | -0.15 |
| VIEScore | CIG-SC | 0.06 | -0.21 | 0.08 | -0.20 | 0.12 | -0.12 | -0.07 | -0.24 | 0.15 | 0.09 | -0.02 | -0.19 | -0.09 | -0.13 |
| | CIG-PQ | 0.01 | -0.81 | 0.01 | -0.82 | 0.03 | -0.64 | -0.01 | -0.60 | 0.04 | -0.40 | 0.03 | -0.63 | 0.02 | -0.46 |
| | MIE-SC | 0.03 | -0.45 | 0.05 | -0.41 | 0.09 | -0.14 | -0.04 | -0.35 | 0.06 | -0.35 | -0.02 | -0.41 | -0.00 | -0.17 |
| | MIE-PQ | -0.00 | -0.69 | 0.01 | -0.73 | 0.02 | -0.48 | -0.02 | -0.39 | 0.00 | -0.71 | -0.01 | -0.74 | 0.01 | -0.72 |
| | TIE-SC | 0.01 | -0.03 | -0.14 | -0.18 | -0.10 | -0.14 | 0.01 | 0.02 | -0.01 | -0.02 | -0.02 | -0.03 | -0.09 | -0.09 |
| | TIE-PQ | 0.01 | -0.12 | 0.05 | -0.22 | -0.09 | -0.13 | 0.00 | -0.00 | 0.03 | -0.17 | 0.02 | -0.22 | 0.00 | -0.40 |

## G. Stability of Diagnostic Estimates

### G.1. Bootstrap Standard Errors

We additionally evaluate the stability of the proposed diagnostics using bootstrap resampling. As shown in Table 10, the resulting standard errors remain consistently small across benchmarks and models, suggesting that the observed differences in $C_V$ and $\rho$ are robust to sampling variability.

*Table 10.* Phase 1 intrinsic consistency with bootstrap standard errors. $C_V$ denotes prompt consistency (lower is better); $\rho$ denotes marginal reliability (higher is better). Subscripts indicate bootstrap standard error ($B = 1000$).

| Benchmark | Criterion | Gemini-2.5 | | GPT-4o | | GPT-4o-mini | | Qwen3-30b | |
|---|---|---|---|---|---|---|---|---|---|
| | | $C_V$ | $\rho$ | $C_V$ | $\rho$ | $C_V$ | $\rho$ | $C_V$ | $\rho$ |
| SummEval | Relevance | $0.25_{\pm0.002}$ | $0.91_{\pm0.0001}$ | $0.05_{\pm0.001}$ | $0.92_{\pm0.0001}$ | $0.04_{\pm0.001}$ | $0.93_{\pm0.0001}$ | $0.17_{\pm0.001}$ | $0.86_{\pm0.0001}$ |
| | Consistency | $0.07_{\pm0.001}$ | $0.55_{\pm0.0004}$ | $0.05_{\pm0.001}$ | $0.91_{\pm0.0001}$ | $0.92_{\pm0.010}$ | $0.88_{\pm0.0001}$ | $0.15_{\pm0.001}$ | $0.60_{\pm0.0003}$ |
| | Fluency | $0.21_{\pm0.002}$ | $0.89_{\pm0.0001}$ | $0.06_{\pm0.001}$ | $0.90_{\pm0.0001}$ | $0.60_{\pm0.007}$ | $0.89_{\pm0.0001}$ | $0.15_{\pm0.001}$ | $0.89_{\pm0.0001}$ |
| | Coherence | $0.09_{\pm0.001}$ | $0.89_{\pm0.0001}$ | $0.29_{\pm0.004}$ | $0.94_{\pm0.0001}$ | $0.06_{\pm0.001}$ | $0.92_{\pm0.0001}$ | $0.22_{\pm0.002}$ | $0.87_{\pm0.0001}$ |
| TopicalChat | Understandability | $0.18_{\pm0.003}$ | $0.39_{\pm0.0011}$ | $0.20_{\pm0.002}$ | $0.49_{\pm0.0009}$ | $0.16_{\pm0.002}$ | $0.48_{\pm0.0009}$ | $0.03_{\pm0.001}$ | $0.53_{\pm0.0008}$ |
| | Naturalness | $0.29_{\pm0.001}$ | $0.85_{\pm0.0002}$ | $0.11_{\pm0.002}$ | $0.80_{\pm0.0004}$ | $0.28_{\pm0.002}$ | $0.57_{\pm0.0010}$ | $0.18_{\pm0.001}$ | $0.87_{\pm0.0002}$ |
| | Coherence | $0.27_{\pm0.002}$ | $0.79_{\pm0.0002}$ | $0.17_{\pm0.002}$ | $0.81_{\pm0.0003}$ | $0.14_{\pm0.001}$ | $0.84_{\pm0.0003}$ | $0.13_{\pm0.001}$ | $0.85_{\pm0.0002}$ |
| | Engagingness | $0.15_{\pm0.002}$ | $0.78_{\pm0.0002}$ | $0.20_{\pm0.003}$ | $0.79_{\pm0.0003}$ | $0.52_{\pm0.005}$ | $0.72_{\pm0.0004}$ | $0.21_{\pm0.002}$ | $0.87_{\pm0.0002}$ |
| | Groundedness | $0.17_{\pm0.001}$ | $0.72_{\pm0.0002}$ | $0.12_{\pm0.001}$ | $0.70_{\pm0.0002}$ | $0.07_{\pm0.001}$ | $0.71_{\pm0.0001}$ | $0.18_{\pm0.002}$ | $0.73_{\pm0.0001}$ |
| HelpSteer-2 | Helpfulness | $0.03_{\pm0.001}$ | $0.86_{\pm0.0001}$ | $0.08_{\pm0.001}$ | $0.84_{\pm0.0002}$ | $0.30_{\pm0.002}$ | $0.67_{\pm0.0003}$ | $0.27_{\pm0.002}$ | $0.72_{\pm0.0003}$ |
| | Correctness | $0.10_{\pm0.002}$ | $0.72_{\pm0.0002}$ | $0.11_{\pm0.001}$ | $0.76_{\pm0.0002}$ | $0.12_{\pm0.001}$ | $0.60_{\pm0.0004}$ | $0.30_{\pm0.002}$ | $0.68_{\pm0.0003}$ |
| | Coherence | $0.16_{\pm0.002}$ | $0.68_{\pm0.0003}$ | $0.24_{\pm0.003}$ | $0.67_{\pm0.0003}$ | $0.40_{\pm0.003}$ | $0.54_{\pm0.0005}$ | $0.21_{\pm0.002}$ | $0.58_{\pm0.0004}$ |
| | Complexity | $1.08_{\pm0.003}$ | $0.87_{\pm0.0002}$ | $0.16_{\pm0.001}$ | $0.89_{\pm0.0001}$ | $0.30_{\pm0.002}$ | $0.81_{\pm0.0002}$ | $0.39_{\pm0.003}$ | $0.88_{\pm0.0001}$ |
| | Verbosity | $0.17_{\pm0.002}$ | $0.87_{\pm0.0002}$ | $0.04_{\pm0.001}$ | $0.83_{\pm0.0002}$ | $0.21_{\pm0.002}$ | $0.74_{\pm0.0004}$ | $0.20_{\pm0.003}$ | $0.76_{\pm0.0002}$ |
| VIEScore | CIG-SC | $1.11_{\pm0.003}$ | $0.94_{\pm0.0002}$ | $0.30_{\pm0.004}$ | $0.96_{\pm0.0001}$ | $0.82_{\pm0.003}$ | $0.93_{\pm0.0002}$ | $0.62_{\pm0.004}$ | $0.94_{\pm0.0002}$ |
| | CIG-PQ | $1.32_{\pm0.006}$ | $0.94_{\pm0.0002}$ | $0.46_{\pm0.002}$ | $0.93_{\pm0.0002}$ | $0.56_{\pm0.003}$ | $0.93_{\pm0.0002}$ | $0.32_{\pm0.004}$ | $0.94_{\pm0.0002}$ |
| | MIE-SC | $1.01_{\pm0.002}$ | $0.94_{\pm0.0002}$ | $0.52_{\pm0.004}$ | $0.94_{\pm0.0002}$ | $0.37_{\pm0.005}$ | $0.95_{\pm0.0002}$ | $0.45_{\pm0.003}$ | $0.94_{\pm0.0002}$ |
| | MIE-PQ | $1.14_{\pm0.002}$ | $0.92_{\pm0.0002}$ | $0.58_{\pm0.005}$ | $0.93_{\pm0.0002}$ | $1.02_{\pm0.007}$ | $0.94_{\pm0.0002}$ | $0.40_{\pm0.003}$ | $0.94_{\pm0.0002}$ |
| | TIE-SC | $1.00_{\pm0.003}$ | $0.94_{\pm0.0002}$ | $0.32_{\pm0.003}$ | $0.94_{\pm0.0002}$ | $1.11_{\pm0.009}$ | $0.95_{\pm0.0002}$ | $0.64_{\pm0.005}$ | $0.93_{\pm0.0002}$ |
| | TIE-PQ | $1.14_{\pm0.002}$ | $0.93_{\pm0.0002}$ | $0.68_{\pm0.006}$ | $0.92_{\pm0.0002}$ | $0.30_{\pm0.005}$ | $0.93_{\pm0.0002}$ | $0.16_{\pm0.003}$ | $0.92_{\pm0.0003}$ |

| Benchmark | Criterion | Qwen3-235b | | Llama-4-m | | Llama-4-s | |
|---|---|---|---|---|---|---|---|
| | | $C_V$ | $\rho$ | $C_V$ | $\rho$ | $C_V$ | $\rho$ |
| SummEval | Relevance | $0.09_{\pm0.001}$ | $0.89_{\pm0.0001}$ | $0.36_{\pm0.004}$ | $0.83_{\pm0.0002}$ | $0.17_{\pm0.002}$ | $0.92_{\pm0.0001}$ |
| | Consistency | $0.08_{\pm0.001}$ | $0.66_{\pm0.0003}$ | $0.25_{\pm0.002}$ | $0.63_{\pm0.0003}$ | $0.07_{\pm0.001}$ | $0.78_{\pm0.0002}$ |
| | Fluency | $0.09_{\pm0.001}$ | $0.90_{\pm0.0001}$ | $0.13_{\pm0.002}$ | $0.81_{\pm0.0002}$ | $0.15_{\pm0.001}$ | $0.92_{\pm0.0001}$ |
| | Coherence | $0.16_{\pm0.001}$ | $0.92_{\pm0.0001}$ | $0.18_{\pm0.001}$ | $0.84_{\pm0.0001}$ | $0.15_{\pm0.003}$ | $0.88_{\pm0.0001}$ |
| TopicalChat | Understandability | $0.27_{\pm0.004}$ | $0.34_{\pm0.0013}$ | $0.48_{\pm0.005}$ | $0.43_{\pm0.0010}$ | $0.21_{\pm0.002}$ | $0.46_{\pm0.0009}$ |
| | Naturalness | $0.23_{\pm0.002}$ | $0.87_{\pm0.0002}$ | $0.33_{\pm0.003}$ | $0.81_{\pm0.0003}$ | $0.49_{\pm0.006}$ | $0.71_{\pm0.0006}$ |
| | Coherence | $0.17_{\pm0.002}$ | $0.86_{\pm0.0002}$ | $0.33_{\pm0.002}$ | $0.82_{\pm0.0003}$ | $0.25_{\pm0.003}$ | $0.81_{\pm0.0002}$ |
| | Engagingness | $0.16_{\pm0.001}$ | $0.86_{\pm0.0002}$ | $0.11_{\pm0.001}$ | $0.80_{\pm0.0004}$ | $0.21_{\pm0.003}$ | $0.70_{\pm0.0005}$ |
| | Groundedness | $0.30_{\pm0.002}$ | $0.71_{\pm0.0001}$ | $0.18_{\pm0.002}$ | $0.73_{\pm0.0001}$ | $0.10_{\pm0.001}$ | $0.71_{\pm0.0002}$ |
| HelpSteer-2 | Helpfulness | $0.37_{\pm0.002}$ | $0.78_{\pm0.0002}$ | $0.13_{\pm0.002}$ | $0.64_{\pm0.0004}$ | $0.28_{\pm0.002}$ | $0.66_{\pm0.0003}$ |
| | Correctness | $0.25_{\pm0.002}$ | $0.76_{\pm0.0002}$ | $0.24_{\pm0.002}$ | $0.54_{\pm0.0005}$ | $0.28_{\pm0.002}$ | $0.54_{\pm0.0005}$ |
| | Coherence | $0.78_{\pm0.007}$ | $0.60_{\pm0.0004}$ | $0.25_{\pm0.003}$ | $0.40_{\pm0.0006}$ | $0.31_{\pm0.003}$ | $0.49_{\pm0.0005}$ |
| | Complexity | $0.32_{\pm0.001}$ | $0.89_{\pm0.0001}$ | $0.26_{\pm0.002}$ | $0.77_{\pm0.0003}$ | $0.31_{\pm0.001}$ | $0.85_{\pm0.0002}$ |
| | Verbosity | $0.74_{\pm0.007}$ | $0.76_{\pm0.0002}$ | $0.33_{\pm0.004}$ | $0.75_{\pm0.0003}$ | $0.45_{\pm0.004}$ | $0.83_{\pm0.0002}$ |
| VIEScore | CIG-SC | $0.47_{\pm0.006}$ | $0.94_{\pm0.0002}$ | $0.17_{\pm0.004}$ | $0.92_{\pm0.0003}$ | $0.46_{\pm0.004}$ | $0.83_{\pm0.0005}$ |
| | CIG-PQ | $0.37_{\pm0.005}$ | $0.92_{\pm0.0002}$ | $0.36_{\pm0.002}$ | $0.90_{\pm0.0003}$ | $0.29_{\pm0.003}$ | $0.81_{\pm0.0005}$ |
| | MIE-SC | $0.94_{\pm0.006}$ | $0.95_{\pm0.0002}$ | $0.54_{\pm0.004}$ | $0.93_{\pm0.0002}$ | $0.55_{\pm0.003}$ | $0.87_{\pm0.0003}$ |
| | MIE-PQ | $0.61_{\pm0.008}$ | $0.90_{\pm0.0002}$ | $0.32_{\pm0.004}$ | $0.89_{\pm0.0003}$ | $0.58_{\pm0.005}$ | $0.80_{\pm0.0005}$ |
| | TIE-SC | $0.43_{\pm0.005}$ | $0.95_{\pm0.0002}$ | $0.56_{\pm0.004}$ | $0.93_{\pm0.0002}$ | $0.80_{\pm0.006}$ | $0.88_{\pm0.0003}$ |
| | TIE-PQ | $0.67_{\pm0.005}$ | $0.91_{\pm0.0002}$ | $0.84_{\pm0.006}$ | $0.90_{\pm0.0002}$ | $0.77_{\pm0.007}$ | $0.80_{\pm0.0005}$ |

## G.2. Cross-Model Significance Tests

We further conduct Kruskal–Wallis tests across models for each benchmark–criterion pair. Table 11 shows statistically significant differences across models for both $C_V$ and $\rho$ in all settings ($p < .001$).

*Table 11.* Kruskal-Wallis test for model differences ($B$=1000).

| Benchmark | Criterion | $C_V$ | | $\rho$ | |
|---|---|---|---|---|---|
| | | $H$ | | $H$ | |
| SummEval | Relevance | 5663.6 | *** | 6816.2 | *** |
| | Consistency | 5145.9 | *** | 6837.1 | *** |
| | Fluency | 4136.0 | *** | 6355.0 | *** |
| | Coherence | 4103.7 | *** | 6724.9 | *** |
| TopicalChat | Understandability | 4019.9 | *** | 5508.9 | *** |
| | Naturalness | 4112.1 | *** | 6645.2 | *** |
| | Coherence | 4019.6 | *** | 6171.5 | *** |
| | Engagingness | 3441.0 | *** | 6406.8 | *** |
| | Groundedness | 4484.7 | *** | 5484.7 | *** |
| HelpSteer-2 | Helpfulness | 5529.7 | *** | 6702.3 | *** |
| | Correctness | 4701.4 | *** | 6617.4 | *** |
| | Coherence | 4190.3 | *** | 6709.5 | *** |
| | Complexity | 4960.7 | *** | 6618.5 | *** |
| | Verbosity | 5270.2 | *** | 6202.0 | *** |
| VIEScore | CIG-SC | 5711.3 | *** | 5749.9 | *** |
| | CIG-PQ | 4326.0 | *** | 5667.8 | *** |
| | MIE-SC | 4590.6 | *** | 5476.2 | *** |
| | MIE-PQ | 4809.0 | *** | 6168.0 | *** |
| | TIE-SC | 5176.3 | *** | 5351.7 | *** |
| | TIE-PQ | 5236.1 | *** | 5398.2 | *** |

