# OpenReview forum: "Diagnosing the Reliability of LLM-as-a-Judge via Item Response Theory"
_ICML.cc/2026/Conference — ICML 2026 regular_

### Official Review · Reviewer_kDpD · 2026-03-13

**Soundness:** 3
**Presentation:** 3
**Significance:** 3
**Originality:** 3
**Overall Recommendation:** 3
**Confidence:** 3

**Summary:**

This paper studies the reliability of LLM-as-a-Judge, an increasingly important issue as LLM-based evaluation is widely used in benchmark pipelines. The authors propose a diagnostic framework based on Item Response Theory  to assess reliability from two perspectives: intrinsic consistency under prompt perturbations and human alignment with human judgments. The framework is evaluated across multiple judge models and several NLP and vision-language evaluation tasks.

**Compliance With Llm Reviewing Policy:**

Affirmed.

**Final Justification:**

I appreciate the authors' willingness to add practical recommendations. However, since these changes are promised for the revision rather than demonstrated in the current manuscript, I prefer to maintain my score at this time.

**Key Questions For Authors:**

1. Can the framework be extended to pairwise evaluation settings, which are commonly used in RLHF and LLM benchmarking?
2. Are there practical recommendations for improving LLM judge reliability based on the diagnostic results?
3. How does the computational cost scale for large evaluation datasets, and how does it compare with existing evaluation methods?
4. The framework assumes that prompt perturbations (e.g., typos, newlines, paraphrasing) preserve the evaluation semantics. How do the authors ensure that these perturbations do not change the evaluation criteria? In particular, could paraphrasing key terms alter the interpretation of the task?

**Limitations:**

yes

**Strengths And Weaknesses:**

Strengths

1. The paper studies the reliability of LLM-as-a-Judge, which is an important and timely problem as LLM-based evaluation is increasingly used in benchmark evaluation pipelines.
2. The authors evaluate the framework across multiple judge models, multiple NLP tasks, and also include vision-language evaluation settings.
3. The paper reports several useful empirical findings, including prompt sensitivity of LLM judges, greater instability in vision-language evaluation, and systematic exaggeration of quality differences relative to human ratings.

Weaknesses

1. The proposed framework currently only applies to rating-based evaluation and cannot directly handle pairwise or open-ended evaluation settings, which limits its general applicability.
2. The paper introduces IRT-based reliability metrics but does not systematically compare them with standard reliability measures such as Cohen’s κ or Krippendorff’s α.
3. Although the paper reports strong modality-dependent differences, it provides limited analysis of why vision-language evaluation is especially sensitive to prompt variations.

---

> ### Author Rebuttal · Authors · 2026-03-30
>
> Dear Review kDpD,
>
> We sincerely thank Reviewer kDpD for taking the time to carefully read our paper despite your busy schedule. We are grateful for the recognition of our contributions, and the thoughtful feedback has helped us strengthen the paper substantially. Below, we address each concern in detail, and we would greatly appreciate it if you could review our responses.
>
> ## W1 & Q1
>
> We agree this is a limitation. Our primary goal was to demonstrate the feasibility of IRT as a diagnostic tool for LLM-as-a-Judge reliability, and we focused on rating-based evaluation as it is the most common use case for LLM-as-a-Judge in practice. Extending to pairwise settings is a meaningful direction—for instance, Bradley-Terry models could serve as an IRT analogue for pairwise comparisons. We will make this future direction explicit in the Limitations.
>
> ## Baseline (W2)
>
> Reviewers noted the lack of comparison with standard reliability metrics (McDonald's ω, Cohen's κ, Krippendorff's α). We conducted additional experiments across all model–benchmark pairs; full results are in our anonymous github [https://anonymous.4open.science/r/ICML-2026-7160-0677/baseline_intrinsic.png, https://anonymous.4open.science/r/ICML-2026-7160-0677/baseline_human_alignment.png].
> Phase 1. We initially omitted ω because it captures a different aspect of reliability than CV/ρ. The additional results confirm this: ω ranges 0.79–0.99 with minimal variation, whereas CV (0.03–1.32) and ρ (0.34–0.96) clearly identify problematic pairs. For instance, Gemini-2.5 on VIEScore shows CV > 1.0 (severe prompt instability) yet ω > 0.91 — ω preserves relative ordering but cannot detect within-rating instability (CV) or the true-variance vs. estimation-error decomposition (ρ). We will include ω results and a contrastive discussion in the appendix.
>
> Phase 2. We had reported Pearson r in Appendix E; we will add κ and α from additional results. These confirm the broad pattern (moderate NLP agreement, near-zero vision agreement) but cannot distinguish why alignment fails. Our θ_ratio + score-level θ analysis separates calibration mismatch (NLP: same construct, different scales; potentially fixable) from validity gap (vision: different constructs; not fixable by rescaling). We will add κ/α results and a contrastive discussion in the appendix.
>
> ## W3
>
> We agree this is an important question, but diagnosing the internal mechanisms behind VLM prompt sensitivity would require analyzing model internals (e.g., attention patterns, cross-modal representations), which is beyond the scope of our diagnostic framework. Our framework is a new tool that identifies that vision tasks are more prompt-sensitive; explaining why is a separate research question. We will note this as a future direction.
>
> ## Q2
>
> We observe that prompt design choices (e.g., detail, CoT, scale) can affect reliability, but their impact is context-dependent. We therefore recommend using our framework to diagnose and guide adjustments of such variations, rather than adopting them as universally improving strategies. Our primary focus is on enabling diagnosis and interpretation, while improving reliability based on these signals is left as future work.
>
>
> ## Q3
>
> GRM fitting takes approximately 5 minutes per triple of (model, benchmark, criterion) on a single CPU (AMD EPYC 7313). This is higher than simple agreement metrics (e.g., κ, α) but remains practically acceptable as a diagnostic tool.
>
> ## About Prompt Variation (Q4)
>
> Minimal perturbations were a deliberate design choice. If a judge is unstable under trivial surface changes (typos, newlines, synonym choices) that naturally arise during prompt authoring, this raises serious concerns about its reliability. Diagnosing such sensitivity serves as a minimal prerequisite for reliable evaluation. Moreover, our framework is intended for researchers to assess whether their own judge prompt is stable, so perturbations must remain close to the original; otherwise, the result may reflect a different judge altogether, undermining the diagnostic objective.
>
> For broader prompt-level changes such as CoT, detailed instructions, and scale adjustment, we separately examine their effects in the ablation study (Table 2), showing how such design choices systematically shift CV and ρ. This separation is intentional: perturbations diagnose the stability of a fixed judge, while ablations guide how to redesign or improve it.
>
> We will revise Section 3.3 to more clearly articulate this design intent.
>
>
> We are committed to incorporating all the discussed changes in the final version of the manuscript. Thank you again for your valuable feedback.
>
> Best regards,
> The Authors

---

> > ### Author Rebuttal · Reviewer_kDpD · 2026-04-03
> >
> > Thank you for your detailed rebuttal. The baseline comparison (W2) and prompt perturbation rationale (Q4) are convincing. My remaining concern is that the framework currently diagnoses reliability issues but stops short of offering concrete guidance on how practitioners should act on these signals. Translating the diagnostic findings into actionable recommendations would strengthen the paper's practical impact. I maintain my current score.

---

> > > ### Author Response · Authors · 2026-04-04
> > >
> > > Dear Reviewer kDpD,
> > >
> > > Thank you for your continued engagement and for finding our additional baseline comparisons and prompt perturbation rationale convincing.
> > >
> > > Regarding the lack of prescriptive practical guidance: we sincerely acknowledge this limitation. Actually, our ablation study (Table 2) did reveal clear tendencies — for example, detailed instructions tend to reduce CV, and scale adjustment affects ρ. However, as LLM-as-a-judge is utilized across highly diverse settings, we were initially cautious about generalizing these observations into definitive guidelines. We believed that formulating robust, universally applicable recommendations would require significantly more test cases across varying contexts, effectively constituting a separate, independent line of research.
> > >
> > > However, we now realize, as you and Reviewer KCeF have rightfully pointed out, that stopping short of offering practical guidance limits the framework's immediate utility. We completely agree that translating these diagnostic signals into actionable recommendations is crucial for practitioners. Furthermore, we recognize that explicitly summarizing the key findings and empirical "hints" derived from our framework will provide a valuable foundation and direction for future research in this area. Therefore, in the camera-ready version, we will add a dedicated paragraph outlining actionable recommendations between Section 6.3 (Discussion) and Section 7 (Conclusion), carefully framed within the scope of our study, as follows. Also, we will use the extra page to summarize the core findings obtained through our analysis to guide practitioners to adjust their evaluation prompts.
> > >
> > > > Our experiments suggest several tendencies that may help practitioners interpret and act on the diagnostic signals, based on the settings considered in this study. **When prompt sensitivity (CV) is high**, adding more detailed evaluation instructions with explicit rubric definitions tends to improve stability, with additional gains from chain-of-thought prompting (Table 2). **When marginal reliability (ρ) is low or unstable**, adjusting the rating scale may help improve measurement reliability, as different scales yield substantially different ρ values (Table 2). For human alignment, NLP tasks often exhibit patterns consistent with calibration mismatch, suggesting that differences may be partially addressed through post-hoc rescaling. **In VIEScore tasks, judges often exhibit patterns indicative of a construct validity gap, suggesting that they may capture different aspects of quality than humans (Figure 1)**. While our framework identified this tendency, extending the analysis to additional benchmarks remains an important direction for strengthening the generalizability of diagnostic findings.
> > >
> > > We will also state the scope and limitations of these recommendations within a dedicated Limitations section.
> > >
> > > Best regards,
> > >
> > > The Authors

---

### Official Review · Reviewer_KCeF · 2026-03-13

**Soundness:** 2
**Presentation:** 2
**Significance:** 2
**Originality:** 3
**Overall Recommendation:** 4
**Confidence:** 4

**Summary:**

Existing work studies the intrinsic consistency and human alignment of an LLM judge separately. To bridge this gap, the authors innovatively introduce IRT methods and design a two-phase framework to evaluate the LLM judges. By conducting experiments across diverse models and tasks, the authors analyze the patterns revealed by the metrics and discuss potential sources of unreliability.

**Compliance With Llm Reviewing Policy:**

Affirmed.

**Final Justification:**

The clarifications provided have satisfactorily addressed my main concerns. Accordingly, I am increasing my overall recommendation from 3 to 4.

**Key Questions For Authors:**

1. As shown in Weaknesses #2, the proposed method lacks direct comparisons with some related work. Under a more conventional setup, if we separately examine a model’s intrinsic consistency and human alignment, versus using the proposed two-phase framework to assess the same model, what is the fundamental difference? What are the irreplaceable benefits of the proposed approach?

2. Although both phases are computed based on the IRT latent quality $\theta$, intrinsic consistency and human alignment are ultimately evaluated using two different metrics. Is it possible or necessary to propose a unified metric that jointly measures performance across both phases? (Non-critical)

3. Does the proposed method require statistical experiments to discuss its sensitivity or stability with respect to the amount of data (samples or items)?

**Limitations:**

The authors state some of their limitations in Impact Statement.

- The framework is applicable only to point-scale (rating-based) judgments.

- The internal reasoning produced by LLM judges is not examined.

- Multilingual evaluation and other modalities remain unexplored.

Additionally, the authors could improve the paper’s quality by addressing the weaknesses discussed above.

**Strengths And Weaknesses:**

## Strengths

1. Using the IRT method, the intrinsic consistency and human alignment of the LLM judge can be measured based on the latent quality $\theta$, which is an interesting idea.
2. The analysis of the experiments is concrete and solid, including analyses of results across different models, tasks, and modalities.
3. Beyond diagnosing the reliability of LLM judges, the authors also attempt a deeper discussion and derive several interesting inferences, such as `VLMs may evaluate different aspects of quality than what humans evaluate`.

## Weaknesses

1. Lack of clarity: The structure and organization of this paper are somewhat unconventional. For example, Section 5 and Section 6 are essentially experimental results, yet similar titles already appear in Section 3.4 to describe the framework and metrics. In addition, Section 5 and Section 6 contain many findings (e.g., differences in metric values across models, tasks, and modalities), but the paper lacks ways to better summarize or highlight the underlying causes and the most interesting or counter-intuitive insights as key contributions.

2. There are many related works on both intrinsic consistency and human alignment. Although the authors adopt an IRT approach, this paper does not directly compare each phase against relevant baselines, making it unclear what improvements the IRT method brings to Phase 1 and Phase 2.

3. There are also many IRT algorithms in that field. The paper gives motivations for the choice of IRT-GRM and corresponding settings, but does not provide empirical comparisons to demonstrate the advantages of this specific choice.

minor:

1. The IRT-GRM fitting configuration is provided, but the paper does not clearly describe the fitting setup (i.e., whether, for any given judge model and benchmark, the IRT-GRM is fit once using ratings from all prompt variations).

typo error:

1. There is an incorrect footnote reference in the last sentence of the abstract.

2. In line 179, a space is missing between `judge.` and `For`.

3. In section 4.3, there may be a misuse of "phrase" (should be "paraphrase").

---

> ### Author Rebuttal · Authors · 2026-03-30
>
> Dear Review KCeF,
>
> We sincerely thank Reviewer KCeF for taking the time to carefully read our paper despite your busy schedule. We are grateful for the recognition of our contributions, and the thoughtful feedback has helped us strengthen the paper substantially. Below, we address each concern in detail, and we would greatly appreciate it if you could review our responses.
>
>
> ## Lack of clarity (W1)
>
> Thank you for helping improve the clarity. We will rename Sections 5–6 to "Results: Intrinsic Consistency" and "Results: Human Alignment" to distinguish them from the metric definitions in Section 3.4. We will also add a concise summary of key findings at the beginning of each Discussion subsection (5.3, 6.3).
>
>
> ## Baseline (W2 & Q1)
>
> Reviewers noted the lack of comparison with standard reliability metrics (McDonald's ω, Cohen's κ, Krippendorff's α). We conducted additional experiments across all model–benchmark pairs; full results are in our anonymous github [https://anonymous.4open.science/r/ICML-2026-7160-0677/baseline_intrinsic.png, https://anonymous.4open.science/r/ICML-2026-7160-0677/baseline_human_alignment.png].
> Phase 1. We initially omitted ω because it captures a different aspect of reliability than CV/ρ. The additional results confirm this: ω ranges 0.79–0.99 with minimal variation, whereas CV (0.03–1.32) and ρ (0.34–0.96) clearly identify problematic pairs. For instance, Gemini-2.5 on VIEScore shows CV > 1.0 (severe prompt instability) yet ω > 0.91 — ω preserves relative ordering but cannot detect within-rating instability (CV) or the true-variance vs. estimation-error decomposition (ρ). We will include ω results and a contrastive discussion in the appendix.
>
> Phase 2. We had reported Pearson r in Appendix E; we will add κ and α from additional results. These confirm the broad pattern (moderate NLP agreement, near-zero vision agreement) but cannot distinguish why alignment fails. Our θ_ratio + score-level θ analysis separates calibration mismatch (NLP: same construct, different scales; potentially fixable) from validity gap (vision: different constructs; not fixable by rescaling). We will add κ/α results and a contrastive discussion in the appendix.
>
>
> ## IRT Model Choice (W3)
>
> GRM is the standard IRT model for ordered categorical responses, which exactly matches our setting (Likert-scale ratings). As we want to follow statistical conventions strictly, we selected it. Other IRT models are not applicable in that sense. They are designed for different data types: NRM assumes nominal (unordered) categories, and 2PL handles only binary responses—which we already apply for binary criteria as noted in Section 3.1. The Partial Credit Model (PCM) is another alternative for ordered data but assumes uniform discrimination across items, which may be restrictive in our setting where prompts can exhibit varying sensitivity. We will clarify this justification in Section 3.1. And, we want to note that this is the first attempt to adopt IRT for assessing judgement, and further research could be done based on ours.
>
>
> ## GRM Fitting Setup (Minor1)
>
> The fitting procedure is described in Lines 130–135 (Section 3.1): for each judge model and benchmark criterion, we fit GRM once using ratings from all prompt variants simultaneously, where each variant has its own (αp, βp) while sharing θj across variants. We understand that the manuscript may not clearly deliver the procedure. Thus, we will elaborate on the description to make the procedure clearer.
>
> ## Typo
>
> We appreciate the careful reading. We will fix all noted typos and thoroughly proofread the manuscript in the final version.
>
> ## About Unified Metric (Q2)
>
> We intentionally separate the two phases since they capture different aspects of reliability (measurement stability vs. human alignment). A unified metric could be dominated by one component and obscure the source of failure, making diagnosis less interpretable. We therefore prefer to keep them separate.
>
> ## Stability of Metric Estimates (Q3)
> To address concerns regarding sampling stability and statistical significance, we computed bootstrap standard errors (SE) (B=1000) for all CV and ρ estimates. The SEs are consistently minimal across all benchmarks (CV $\le$ 0.010; ρ $\le$ 0.0013), demonstrating that our diagnostic signals are robust to sampling variability. For instance, TopicalChat-Understandability consistently yields low ρ (0.34–0.53) with SE $\le$ 0.0013, confirming that its poor reliability is a systemic task-level property rather than a sampling artifact. These results prove that the reported differences between models and modalities are statistically significant. We will add the full SE table to the appendix. [https://anonymous.4open.science/r/ICML-2026-7160-0677/Stability_MetricEstimates.png]
>
> We are committed to incorporating all the discussed changes in the final version of the manuscript. Thank you again for your valuable feedback.
>
> Best regards,
> The Authors

---

> > ### Author Rebuttal · Reviewer_KCeF · 2026-04-02
> >
> > Thank you for the detailed response. I now have a clearer understanding that the two phases proposed in the paper and their corresponding related works (or baselines) do not focus on exactly the same questions, and that the proposed method is intended to provide a more fine-grained analysis.
> >
> > However, I still believe the paper has an important limitation: the diagnostic and interpretive value of the proposed method is currently reflected mainly in post hoc analyses of the experimental results and in the discussion of the method, rather than in showing how these diagnostic signals could be used to make more practical evaluation decisions (e.g., through case studies). In addition, the paper currently feels somewhat rushed.
> >
> > Considering the above, I will maintain my score.

---

> > > ### Author Response · Authors · 2026-04-03
> > >
> > > Dear Reviewer KCeF
> > >
> > > Thank you for your continued engagement and for acknowledging that our proposed framework provides a more fine-grained analysis compared to existing baselines.
> > >
> > > Regarding the concern that our method’s value is reflected mainly in post hoc analyses: we respectfully note that diagnosing judge reliability is inherently a post hoc task. A judge’s reliability can only be assessed after it produces judgments — there is no way to evaluate measurement stability before observing the measurements themselves. This is not specific to our framework; existing metrics (ω, κ, α) are equally post hoc. In fact, we view this as a strength: because our framework requires only observed ratings, it is model-agnostic and can be applied to any judge, including commercial API models (e.g., GPT-4o, Gemini) where internal access is unavailable. This generality would not be possible with a non-post-hoc approach.Regarding the lack of guidance on making practical evaluation decisions: we sincerely acknowledge this limitation. Our ablation study (Table 2) did reveal clear tendencies — for example, detailed instructions tend to reduce $C_V$, and scale adjustment affects $\rho$. However, as LLM-as-a-judge is utilized across highly diverse settings, we were initially cautious about generalizing these observations into definitive guidelines. We believed that formulating robust, universally applicable recommendations would require significantly more test cases across varying contexts, effectively constituting a separate, independent line of research.
> > >
> > > We now realize, as you and Reviewer kDpD have rightfully pointed out, that stopping short of offering practical guidance limits the framework's immediate utility. We completely agree that demonstrating how these diagnostic signals can inform practical evaluation decisions is crucial for practitioners. Furthermore, we recognize that explicitly summarizing the key findings and empirical "hints" derived from our framework will provide a valuable foundation and direction for future research in this area. Therefore, in the camera-ready version, we will add a dedicated paragraph outlining actionable recommendations between Section 6.3 (Discussion) and Section 7 (Conclusion), carefully framed within the scope of our study, as follows. Also, we will use the extra page to summarize the core findings obtained through our analysis to guide practitioners to adjust their evaluation prompts.
> > >
> > > > Our experiments suggest several tendencies that may help practitioners interpret and act on the diagnostic signals, based on the settings considered in this study. **When prompt sensitivity (CV) is high**, adding more detailed evaluation instructions with explicit rubric definitions tends to improve stability, with additional gains from chain-of-thought prompting (Table 2). **When marginal reliability (ρ) is low or unstable**, adjusting the rating scale may help improve measurement reliability, as different scales yield substantially different ρ values (Table 2). For human alignment, NLP tasks often exhibit patterns consistent with calibration mismatch, suggesting that differences may be partially addressed through post-hoc rescaling. **In VIEScore tasks, judges often exhibit patterns indicative of a construct validity gap, suggesting that they may capture different aspects of quality than humans (Figure 1)**. While our framework identified this tendency, extending the analysis to additional benchmarks remains an important direction for strengthening the generalizability of diagnostic findings.
> > >
> > >
> > > Regarding the writing quality: the revisions promised in our rebuttal (W1, W4–W11) will be fully incorporated in the camera-ready version, along with thorough proofreading to ensure clarity and coherence throughout the manuscript.
> > >
> > > Best regards,
> > >
> > > The Authors

---

### Official Review · Reviewer_C1jX · 2026-03-13

**Soundness:** 2
**Presentation:** 3
**Significance:** 2
**Originality:** 2
**Overall Recommendation:** 4
**Confidence:** 4

**Summary:**

This paper proposes a two-phase diagnostic framework for evaluating the reliability of LLM-as-a-Judge approaches using item response theory. Phase 1 assesses intrinsic consistency via stability under prompt variation, while Phase 2, conditional on passing Phase 1, evaluates  alignment with human . The framework is applied across seven judge models on four benchmarks for NLP and vision tasks.

**Compliance With Llm Reviewing Policy:**

Affirmed.

**Final Justification:**

I am still skeptical of the practical relevance given the minimum tested variations in phase 1 but my other concerns were largely resolved. I have raised my score to reflect this.

**Key Questions For Authors:**

1. The GRM separates prompt effects from true quality differences, but assumes these are the only two sources of variation. How does the model account for other known reliability concerns such as self-preference bias (where LLM judges favor outputs from the same model family)?
2. What is the justification/concrete reference for the CV < 0.1 threshold for acceptable consistency? The same question applies to rho > 0.7. How is the threshold determined for intrinsic consistency to advance it to phase 2?
3. The prompt variations tested are minimal surface-level perturbations. Would more substantial semantic variations (e.g., different evaluation framing, reordered rubric items) still preserve the assumption that the same latent quality is being measured? If not, where is the boundary, and how should practitioners think about it?
4. The paper uses the term 'validity gap' to describe the vision task findings. Validity is a precise psychometric term with multiple subtypes (construct, content, criterion, etc.). Which type of validity is being referred to here, and is 'validity gap' the appropriate framing? (see this paper that describes different validity types in the context of AI evaluations: https://arxiv.org/pdf/2505.10573; or what kind of validity is meant here?)
5. What was the procedure for character- level perturbations?

**Limitations:**

The main body of the paper has no dedicated limitations section, and the conclusion does not mention limitations at all. Assumptions aren't discussed at all. Please add a dedicated section discussing both limitations and assumptions (see some of the weaknesses mentioned above as examples on what to add).

**Strengths And Weaknesses:**

**Strengths**
- The core technical approach is well-grounded. Adapting IRT to LLM-as-a-judge evaluations is a reasonable methodological choice.
- The paper is clearly structured. The two-phase framework maps well onto the experimental sections, and the use of three analytical axes (modality, model, task) is consistent and systematic.
- LLM-as-a-judge approaches are widely deployed but poorly understood as a measurement instrument. A principled diagnostic framework with interpretable metrics addresses a real gap.
- The framework is restricted to rating-based (point-scale) judgments and does not extend to pairwise comparisons or open-ended evaluations, which limits applicability to common LLM-as-a-Judge setups such as MT-Bench or Chatbot Arena.


**Weaknesses**
- The threshold of C_V < 0.1 for acceptable consistency is asserted without justification. The paper does not cite a precedent or provide a derivation for this cutoff. The same applies to rho > 0.7, where a source for the presumed psychometric convention is mentioned but not cited.
- The prompt variation set is narrow (typos, newlines, and synonym substitution). These are minimal surface perturbations. More semantically varied prompts (e.g., different evaluation framing or instruction ordering) might reveal additional inconsistencies. The paper does not discuss whether expanding variations would compromise the assumption that the same latent quality is being measured.
-  The generalization claims about VLMs are based on a single benchmark, which isn't sufficient evidence to make statements about VLMs more broadly..
- In the diagnostic interpretation sections, not all potential combinations are explained (e.g., in line 188/189, interpretation guidelines for the case low C_V, low \rho is missing. While they may seem trivial, this would help practitioners less familiar with the material to adopt your framework.
- (Minor) Citation for IRT is missing (e.g., reference IRT Handbook upon first mention)
- (Minor) Several acronyms are used before being spelled out (e.g., NUTS in Section 3.1). This should be fixed.
- (Minor) The abstract uses the vague term 'diverse LLM judges' instead of stating the number of models evaluated. This should be made concrete.
- (Minor) The term 'intrinsic reliability' is used in the related work section without having been formally introduced at that point; the paper introduces 'intrinsic consistency' in the abstract. The terminology should be consistent.
- (Minor) The ablation study is introduced and discussed in Section 5.3 without prior methodological setup. It should be described in Section 3 or 4.
- The discussion in Section 5.3 refers to 'scale law' based on two data points per model family. This seems to me too strong a claim for such limited evidence.
- (Minor) Several typos are present (e.g., lines 356, 357, 364). The sentence in Appendix F ('Refer to Figure 2 to') is incomplete.

---

> ### Author Rebuttal · Authors · 2026-03-30
>
> Dear Review C1jX,
>
> We sincerely thank Reviewer C1jX for taking the time to carefully read our paper despite your busy schedule. We are grateful for the recognition of our contributions, and the thoughtful feedback has helped us strengthen the paper substantially. Below, we address each concern in detail, and we would greatly appreciate it if you could review our responses.
>
> ## Justification for Thresholds (W1 & Q2)
>
> We select the thresholds for CV based on the following three established guidelines to ensure the stability of the diagnostic framework:
>
> 1) Statistical Robustness:
> We thought that CV = 0.1 admits an intuitive interpretation under any probabilistic distribution: Since CV = 𝜎/𝜇, a deviation of 2𝜎 corresponds to approximately ±20% of the mean. By Chebyshev’s inequality, the probability of deviations exceeding 2𝜎 is at most 25%, implying that at least 75% of assessments lie within this range even under worst-case distributions. This establishes CV < 0.1 as a conservative and distribution-agnostic criterion for limiting stochastic variability induced by prompt perturbations.
>
> 2) Signal-to-Noise Interpretation:
> In linear signal processing, CV can be interpreted as the reciprocal of the Signal-to-Noise Ratio (SNR = 1/CV). Thus, CV < 0.1 corresponds to a regime where signal dominates variability, meaning that measurement fluctuations remain small relative to the underlying signal. This provides an intuitive interpretation of stability rather than a strict domain-specific guarantee. Note that SNR > 5 is treated as a minimal performance of discriminating signal from noises, and CV < 0.1 corresponds to SNR > 10.
>
> 3) Cross-domain Scientific Convention: A CV below 10% is widely regarded as a benchmark for high measurement precision across experimental sciences. This threshold is established as a rule of thumb in analytical chemistry [1] and clinical epidemiology [2], where CV < 10% is classified as low variability indicating reproducibility. This reflects a common requirement for a specific precision level to support reliable numerical inference.
>
> Additionally, we adopt ρ > 0.7 based on psychometric conventions [3] for acceptable marginal reliability.
>
> We will elaborate on this rationale in Section 3.4 with the following citations.
>
> [1] Reed, G. F., Lynn, F., & Meade, B. D. (2002). Use of Coefficient of Variation in Assessing Variability of Quantitative Assays. Clinical and Diagnostic Laboratory Immunology, 9(6), 1235–1239.
>
> [2] Shechtman, O. (2013). The Coefficient of Variation as an Index of Measurement Reliability. In Methods of Clinical Epidemiology, Springer, pp. 39–49.
>
> [3] Nunnally, J. C. (1978). Psychometric Theory, 2nd ed. McGraw-Hill.
>
> ## About Prompt Variation (W2 & Q3)
> Refer to Reviewer uaEj rebuttal (§ about prompt variation).
>
> ## VLM Generalization (W3)
>
> While VIEScore is a single benchmark, it covers three distinct subtasks (CIG, TIE, MIE) with different evaluation aspects, and our findings were consistent across all three. We believe this internal diversity provides reasonable evidence for our claims, but agree that extending to additional VLM benchmarks would further strengthen generalizability. We will add this as an explicit limitation.
>
> ## Clarification
>
> We will ...
> - W4 Add the missing diagnostic combination in Section 3.4
> - W5 Add IRT citation upon first mention
> - W6 Spell out all acronyms before use (e.g., NUTS)
> - W7 Replace "diverse LLM judges" with the concrete number (seven models) in the abstract
> - W8 Unify terminology ("intrinsic reliability" → "intrinsic consistency" throughout)
> - W9 Move ablation methodology description to Section 4
> - W10 Weaken "scale law" to "scale impacts" (Section 5.3)
> - W11 Fix all noted typos and the incomplete sentence in Appendix F
>
> Also, for Q5: For character-level perturbations, the procedure is described in Appendix B.1
>
> ## Self-Preference Bias (Q1)
> Our current work is the first attempt to apply IRT to LLM-as-a-Judge, focusing on prompt-level variability. Self-preference bias can be naturally incorporated via Differential Item Functioning in future extensions. We will add this discussion to the Limitations section.
>
> ## Term of Validity Gap (Q4)
> We agree that "validity" is a precise term, and our use of "validity gap" is most closely related to construct validity, as it reflects a mismatch between model-inferred quality and human judgments. We will add a footnote to clarify the specific type of validity intended and adjust the terminology to avoid ambiguity.
>
> ## Limitation
>
> We acknowledge that limitations were discussed only in the Impact Statement. We will use the extra page allowed in the final version to add a dedicated Limitations section in the main body, covering the assumptions and scope of our framework.
>
> We are committed to incorporating all the discussed changes in the final version of the manuscript. Thank you again for your valuable feedback.
> Best regards,
> The Authors

---

> > ### Author Rebuttal · Reviewer_C1jX · 2026-04-03
> >
> > Dear authors, thank you for your response. I appreciate the clarification regarding thresholds, which fully resolves that concern. However, my two main concerns remain unresolved.
> >
> > **C1 (Prompt variations):** I read the reply to uaEj. I agree that variations changing semantic meaning constitute a different construct. But there is a wide space of semantic-preserving variations that your framework does not cover. For instance, paraphrasing full sentences rather than just substituting synonyms, reordering rubric items, ... are all meaning-preserving and representative of how prompts actual differ across practitioners (or even for one practitioner) yet go meaningfully beyond the surface-level noise you test. A judge passing Phase 1 under your current perturbations could still be highly unreliable under any of them. As you acknowledge, your perturbation set constitutes only a "minimal prerequisite" (as per your response to uaEj). With such a narrow set of perturbations, the framework cannot, contrary to what the paper title suggests, diagnose the reliability of an LLM-as-a-judge. It can only detect unreliability under a very restricted class of surface-level noise. Either the perturbation set should be broadened, or the claims about what Phase 1 certifies need to be scoped much more tightly throughout the paper, not just in a limitations section.
> >
> > **C2 (VLM generalization)**: Even granting that VIEScore's three subtasks have some diversity, they share the same authors, and hence (I assume, please correct me if I'm wrong) the same prompt structure, rating scales, .... and hence one benchmark with three subtasks is not sufficient to support modality-level claims. The issue is not just about adding a limitations caveat. Statements like "Vision is more prompt-sensitive than NLP" and "Vision achieves high reliability" appear as section-level conclusions and read as general findings about the modality which, as mentioned, I strongly believe are not supported based on your experiments. These claims should, if you decide to keep them, only be scoped to VIEScore throughout the analysis sections, and not just qualified in a separate limitations paragraph.
> >
> > Since both concerns remain, I will maintain my overall recommendation. I do want to acknowledge that the core idea of applying IRT to LLM-as-a-Judge evaluation is promising, and the threshold justification and many of the clarifications offered in the rebuttal are appreciated, but the research in its current form seems premature.

---

> > > ### Author Response · Authors · 2026-04-06
> > >
> > > Dear Reviewer C1jX,
> > >
> > > Thank you for the continued engagement. We appreciate the acknowledgment that applying IRT to LLM-as-a-Judge evaluation is a promising direction.
> > >
> > > Before addressing C1 and C2, we briefly restate what our framework does. The core contribution is an IRT-GRM-based diagnostic methodology that separates measurement properties from true sample quality, producing interpretable diagnostic signals for LLM judge reliability. Existing reliability metrics operate on observed scores and cannot perform this decomposition. This methodological capability is what we contribute, and the framework itself is agnostic to the specific perturbation design or benchmark used; practitioners can apply it in any evaluation setting they choose.
> > >
> > > **Regarding C1**
> > >
> > > We agree that meaning-preserving prompt variations extend beyond our current perturbation set. However, two points are important for contextualizing this concern.
> > >
> > > First, even under minimal perturbations, a substantial number of judge–criterion pairs already fail Phase 1. Existing metrics fail to detect these problems. Our framework reveals them precisely because IRT-GRM decomposes prompt-induced variance from true quality differences. If current LLM judges cannot pass this minimal stability check, resolving this baseline instability is a necessary first step before extending the perturbation scope.
> > >
> > > Second, we argue that full-sentence paraphrasing or structural changes should not be treated as "variations" of a single instrument, even if meaning-preserving for humans. The underlying mechanism of LLMs is fundamentally different from human cognition: meaning-preserving reformulations can substantially alter LLM outputs [1], affect model rankings [2], and prompt sensitivity has been formalized as generalization error from overfitting to specific token sequences [3]. Our ablation (Table 2) corroborates this: simply adding detailed instructions significantly altered CV. If adding detail inherently changes the tool's variance, full paraphrasing would conflate intrinsic unreliability with variance from a new prompt structure. We therefore treat structural reformulation as creating a new instrument, which should undergo Phase 1 independently.
> > >
> > > Phase 1's objective is thus to evaluate the stability of one fixed prompt against surface-level noise, not to compare multiple prompts. This is why paraphrase perturbation was limited to synonym substitution, isolating surface noise from structural alteration.
> > >
> > > We agree that the paper should clearly convey this scope. We will tighten claims in Sections 3.3, 5, and the Conclusion, and explain why structural changes constitute a new instrument at the beginning of Appendix B.
> > >
> > > **Regarding C2**
> > >
> > > It is correct that the three VIEScore subtasks originate from the same authors and share a similar structural template. However, we would like to respectfully clarify the assumption that these subtasks lack diversity. VIEScore utilizes the ImagenHUB, which integrates a wide variety of underlying datasets. Furthermore, the target tasks involve fundamentally different evaluation rubrics. Following the same reasoning as in C1, changing these core instruction tokens creates a different measurement instrument, which is why we observed consistent diagnostic patterns across these constructs — this in itself demonstrates that the IRT-GRM framework produces reliable signals across distinct evaluation settings.
> > >
> > > That said, we fully agree that evaluating VIEScore alone is not sufficient to make definitive conclusions about the entire vision modality. We will replace broad modality-level terms with the specific benchmark name throughout the manuscript (e.g., "VIEScore subtasks exhibit higher prompt sensitivity than the evaluated NLP benchmarks" rather than "Vision is more prompt-sensitive than NLP"). This applies to section-level headings and in-text conclusions in Sections 5 and 6 — not only in a limitations paragraph. We will also append the following clarification at the end of the relevant modality discussion paragraphs:
> > >
> > > > While consistently observed within VIEScore, generalizing these tendencies to the entire vision modality requires further validation across diverse multimodal benchmarks.
> > >
> > > Best regards, The Authors
> > >
> > > [1] Sclar, M., Choi, Y., Tsvetkov, Y., & Suhr, A. (2023). Quantifying Language Models' Sensitivity to Spurious Features in Prompt Design or: How I learned to start worrying about prompt formatting. arXiv preprint arXiv:2310.11324.
> > >
> > > [2] Mizrahi, M., Kaplan, G., Malkin, D., Dror, R., Shahaf, D., & Stanovsky, G. (2024). State of what art? a call for multi-prompt llm evaluation. Transactions of the Association for Computational Linguistics, 12, 933-949.
> > >
> > > [3] Cox, K., Xu, J., Han, Y., Xu, R., Li, T., Hsu, C. Y., … & Ding, Y. (2025, April). Mapping from meaning: Addressing the miscalibration of prompt-sensitive language models. In Proceedings of the AAAI Conference on Artificial Intelligence (Vol. 39, No. 22, pp. 23696-23703).

---

### Official Review · Reviewer_uaEj · 2026-03-16

**Soundness:** 3
**Presentation:** 3
**Significance:** 3
**Originality:** 3
**Overall Recommendation:** 5
**Confidence:** 3

**Summary:**

The paper focuses on diagnosing the reliability of LLM as a judge, by separating measurement properties from the true quality differences. The method proposed uses Graded Response Model (GRM) and it examines two dimensions: (a) intrinsic consistency of LLM as a judge via prompt variations (b) human alignment of the judge via distribution matching.

With their analysis, they are able to show the following: (a) vision tasks seem more prompt sensitive than NLP tasks (b) scale doesn’t guarantee robustness (only helps in NLP tasks) (c) prompt engineering helps (eg: doing chain of thought or adding more details) improve prompt sensitivity (d) NLP tasks show a calibration mismatch with human judgments and vision tasks show validity gaps where judges measure different constructs potentially.

**Compliance With Llm Reviewing Policy:**

Affirmed.

**Key Questions For Authors:**

1. Did you evaluate with more natural paraphrases of the prompt? If not, why?
2. Can you report significant differences in your analysis?
3. How much does the task and the objectivity of evaluation influence the takeaways?
4. How can we use this to predict which tasks / aspects / scale etc will have reliable evaluations with LLM-as-a-judge?
5. Any reasons for why the same aspect  (eg: coherence) for different tasks have very different behaviors?

**Limitations:**

yes

**Strengths And Weaknesses:**

Strengths:
1. Well written paper, well-motivated and the grounding in IRT is nice and novel
2. They test on text and vision tasks + respective frontier judges, showing higher coverage
3. In the framing of the metrics (Section 3) they explain the expected results well
4. They give the inference parameters + prompts for reproducibility
5. Good ablations (eg: checking how the evaluation criteria details and different prompting strategies change CV)
6. The paper identifies actionable causes of unreliability (eg: using detailed prompts)
7. Along with aggregate analysis, the paper also looks into score level analysis to talk about calibration vs validity gaps

Weaknesses:

w1. Prompt variations are limited and local: the paper only tests for local and surface level prompt variations ( typos, adding new lines and changing verbs/synonyms). This doesn’t test for variations which are semantic paraphrases of the global prompt

w2. Missing significance testing to tell if any reported difference between model values is significant or not

---

> ### Author Rebuttal · Authors · 2026-03-30
>
> Dear Review uaEj,
>
> We sincerely thank Reviewer uaEj for taking the time to carefully read our paper despite your busy schedule. We are grateful for the recognition of our contributions, and the thoughtful feedback has helped us strengthen the paper substantially. Below, we address each concern in detail, and we would greatly appreciate it if you could review our responses.
>
> ## About Prompt Variation (W1 & Q1)
>
> Minimal perturbations were a deliberate design choice. If a judge is unstable under trivial surface changes (typos, newlines, synonym choices) that naturally arise during prompt authoring, this raises serious concerns about its reliability. Diagnosing such sensitivity serves as a minimal prerequisite for reliable evaluation. Moreover, our framework is intended for researchers to assess whether their own judge prompt is stable, so perturbations must remain close to the original; otherwise, the result may reflect a different judge altogether, undermining the diagnostic objective.
>
> For broader prompt-level changes such as CoT, detailed instructions, and scale adjustment, we separately examine their effects in the ablation study (Table 2), showing how such design choices systematically shift CV and ρ. This separation is intentional: perturbations diagnose the stability of a fixed judge, while ablations guide how to redesign or improve it.
>
> We will revise Section 3.3 to more clearly articulate this design intent.
>
> ## Significance of Model Differences (W2 & Q2)
> We conducted Kruskal-Wallis tests on CV and ρ across all models for each benchmark-criterion pair [https://anonymous.4open.science/r/ICML-2026-7160-0677/Significance_ModelDifferences.png]. All comparisons yield p < 0.001, confirming that the model differences reported in our analysis are statistically significant. We will add these results to the appendix.
>
> ## Q3 - Q5
> Due to our selection of six evaluation tasks, our results suggest that both task characteristics and the degree of objectivity influence reliability: more subjective or loosely defined aspects tend to exhibit lower consistency and alignment, while well-specified criteria are more stable.
>
> However, our study is limited to a fixed set of benchmarks, and we do not attempt to predict reliability across all tasks or settings. Extending this analysis to more diverse tasks and evaluation setups is an important direction for future work.
>
> For the variation of the same aspect (e.g., coherence), we note that we use the original prompts from each dataset, so the operational definition of an aspect differs across tasks. This indicates that reliability depends not only on the aspect itself, but also on how it is specified in the prompt.
>
>
> We are committed to incorporating all the discussed changes in the final version of the manuscript. Thank you again for your valuable feedback.
>
> Best regards,
> The Authors

---

> > ### Author Rebuttal · Reviewer_uaEj · 2026-04-02
> >
> > Thank you for the authors’ response. I have no further questions.

---

> > > ### Author Response · Authors · 2026-04-06
> > >
> > > Thank you for the thoughtful review and follow-up. We are glad that the concerns raised have been addressed, and we will reflect the discussed points in the final version of the paper.

---

### Decision · Program_Chairs · 2026-04-30

**Decision:**

Accept (regular)

**Comment:**

This paper proposes a two-phase diagnostic framework for evaluating the reliability of LLM-as-a-judge using item response theory.
The separation of intrinsic consistency and human alignment is novel and provides interpretable insights into reliability.
Several concerns are raised by reviewers. The prompt perturbations are limited to surface-level changes. Some claims also appear too general given the limited benchmarks. In addition, the practical implications for decision-making are still somewhat limited.
The rebuttal addressed many technical concerns (e.g., thresholds, significance, baseline comparisons.
Key concerns about perturbation design and practical impact are still not fully resolved, but  overall, the paper presents a promising and valuable contribution despite these limitations.